# Modeling Glacial Lake Outburst Flood Process Chain: The Case of Lake Palcacocha and Huaraz, Peru

**M. A. Somos-Valenzuela[2], R. E. Chisolm[1], D. S. Rivas[1], C. Portocarrero[3] and D. C. McKinney[1]**

[1] {Center for Research in Water Resources, University of Texas at Austin, Austin, Texas, USA}

[2] {Department of Civil and Environmental Engineering, University of Massachusetts, Amherst}

[3] {Instituto Nacional de Investigación en Glaciares y Ecosistemas de Montaña (INAIGEM), Huaraz, Peru}

Correspondence to: D. McKinney (daene@aol.com)

## Abstract

One of the consequences of recent glacier recession in the Cordillera Blanca, Peru, is the risk of Glacial Lake Outburst Floods (GLOFs) from lakes that have formed at the base of retreating glaciers. GLOFs are often triggered by avalanches falling into glacial lakes, initiating a chain of processes that may culminate in significant inundation and destruction downstream. This paper presents simulations of all of the processes involved in a potential GLOF originating from Lake Palcacocha, the source of a previously catastrophic GLOF on December 13, 1941, killing about 1800 people in the city of Huaraz, Peru. The chain of processes simulated here includes: (1) avalanches above the lake; (2) lake dynamics resulting from the avalanche impact, including wave generation, propagation, and run-up across lakes; (3) terminal moraine overtopping and dynamic moraine erosion simulations to determine the possibility of breaching; (4) flood propagation along downstream valleys; and (5) inundation of populated areas. The results of each process feed into simulations of subsequent processes in the chain, finally resulting in estimates of inundation in the city of Huaraz. The results of the inundation simulations were converted into flood intensity and preliminary hazard maps (based on an intensity-likelihood

matrix) that may be useful for city planning and regulation. Three avalanche events with volumes ranging from 0.5-3 x $10^6$ $m^3$ were simulated, and two scenarios of 15 m and 30 m lake lowering were simulated to assess the potential of mitigating the hazard level in Huaraz. For all three avalanche events, three-dimensional hydrodynamic models show large waves generated in the lake from the impact resulting in overtopping of the damming-moraine. Despite very high discharge rates (up to 63.4 x $10^3$ $m^3$ $s^{-1}$), the erosion from the overtopping wave did not result in failure of the damming-moraine when simulated with a hydro-morphodynamic model using excessively conservative soil characteristics that provide very little erosion resistance. With the current lake level, all three avalanche events result in inundation in Huaraz due to wave overtopping, and the resulting preliminary hazard map shows a total affected area of 2.01 $km^2$, most of which is in the high-hazard category. Lowering the lake has the potential to reduce the affected area by up to 35% resulting in a smaller portion of the inundated area in the high-hazard category.

## 1   Introduction

### 1.1   Climate impacts in the Cordillera Blanca of Peru

Atmospheric warming has induced melting of many glaciers around the world (WGMS, 2012; IPCC, 2013; Marzeion et al., 2014). The formation of new lakes in de-glaciating high-mountain regions strongly influences landscape characteristics and represents a significant hazard related to climate change (Frey et al., 2010; Rosenzweig et al., 2007; Kattleman, 2003; Richardson and Reynolds, 2000). The glacier-covered area of the Cordillera Blanca range in Peru has decreased from a Little Ice Age peak of 900 $km^2$ to about 700 $km^2$ in 1970, 528 $km^2$ in 2003, and further decreased to 482 $km^2$ in 2010 (UGRH, 2010; Burns and Nolin, 2014). As a consequence of this glacier recession, many glacial lakes have formed or expanded in the Cordillera Blanca that pose various levels of Glacial Lake Outburst Flood (GLOF) risk for communities below these lakes (Emmer and Vilímek, 2013).

The steep summits of the Cordillera Blanca are undergoing long-term slope destabilization due to warming and permafrost degradation (Haeberli, 2013). Related ice and rock avalanches are especially dangerous in connection with glacial lakes forming or expanding at the foot of steep mountain slopes because they can trigger large waves in the lakes and potentially lead to GLOFs

(Carey et al., 2012; Haeberli, 2013). There are many examples in the Cordillera Blanca of
glacier-related incidents and catastrophes (Lliboutry et al., 1977; Carey, 2010; Portocarrero,
2014). A recent example in the Cordillera Blanca is the 2010 event comprised of a nearly 0.5
million m$^3$ ice/rock avalanche from the summit of Nevado Hualcán that fell into Lake 513 and
generated waves that overtopped the natural rock dam of the lake, producing flood waves and
debris flows that reached the town of Carhuaz (Carey et al., 2012; Schneider et al., 2014).
Preventive lowering of Lake 513 by artificial tunnels in the 1990s, creating a freeboard of 20
meters, helped avoid a major catastrophe that could have killed many people (Reynolds et al.,
1998; Carey et al., 2012; Portocarrero, 2014).

## 10  1.2  Introduction to glacial lake hazard process chain modeling

Emmer and Vilímek (2013, 2014) and Haeberli et al. (2010) have recommended that the
evaluation of glacial lake hazards be based on systematic and scientific analysis of lake types,
moraine dam characteristics, outburst mechanisms, down-valley processes and possible cascades
of processes. Changes in climate patterns are likely to increase the frequency of avalanches as a
consequence of reduced stability of permafrost, bedrock and steep glaciers in the Cordillera
Blanca (Fischer et al., 2012). Under these conditions, avalanches are the most likely potential
trigger of GLOFs (Emmer and Vilímek, 2013; Emmer and Cochachin, 2013; Awal et al., 2010;
Bajracharya et al., 2007; Richardson and Reynolds, 2000; Costa and Schuster, 1988), acting as
the first link in a chain of dependent processes propagating downstream: (1) large avalanche
masses reaching nearby lakes, (2) wave generation, propagation, and runup across lakes, (3)
terminal moraine overtopping and/or moraine breaching, (4) flood propagation along
downstream valleys; and (5) inundation of riverine populated areas (Worni et al., 2014; Westoby
et al., 2014b).
Few studies have attempted to simulate an entire GLOF hazard process chain in a single
modeling environment, generally limiting the number of processes considered; e.g., Worni et al.
(2014) excluded avalanche simulations from their modeling framework. Worni et al. (2014) and
Westoby et al. (2014a) review typical modeling approaches for GLOFs that involve land or ice
masses falling into glacial lakes. An approach that separately simulates individual processes
predominates, where different processes are connected by using the results of one model as the
input for the simulation of the next (e.g. Schneider et al., 2014; Westoby et al., 2014b, Worni et

al., 2014). In this paper, this approach was used to produce simulations of each process in the chain from avalanche to inundation, ensuring that the processes were properly depicted. The glacial lake hazard process chain simulated here includes: avalanche movement into a lake, wave generation and lake hydrodynamics, wave overtopping and moraine erosion, and downstream sediment transport and inundation.

Physical models of avalanche phenomena have been used to simulate the characteristic mass movement processes, e.g., snow avalanches, rock slides, rock avalanches or debris flows (Schneider et al., 2010). Rock-ice avalanches exhibit flow characteristics similar to all of these processes, and the choice of an appropriate model is difficult because available models are not able to fully simulate all of the elements of these complex events. Schneider et al. (2010) tested the Rapid Mass Movements RAMMS model (Bartelt et al., 2013; Christen et al., 2010), a two-dimensional dynamic physical model based on the shallow water equations (SWE) for granular flows and the Voellmy frictional rheology to successfully reproduce the flow and deposition geometry as well as dynamic aspects of large rock-ice avalanches.

Empirical models have been developed that analytically calculate wave characteristics (Heller and Hager, 2010), and some hydrodynamic simulations have been performed for this type of problem (Worni et al., 2012, Schneider et al., 2014). Most hydrodynamic simulations of avalanche-generated waves use two-dimensional models employing the shallow water equations despite the relative importance of vertical accelerations that cannot be considered in 2D shallow water schemes (Heinrich, 1992; Zweifel et al., 2006). Fully 3D, non-hydrostatic models, e.g., FLOW3D (Flow Science, 2012), can simulate the important characteristics of the wave-generation, propagation and overtopping of a terminal moraine in an avalanche-triggered GLOF and take into account irregular lake bathymetries and geometries.

Dynamic modeling of moraine erosion deals with tradeoffs between reliability, complexity, field data demand, and computational power. Several physical processes converge when natural or artificial dams fail; hydrodynamic, erosive, and sediment transport phenomena, as well as movement of boulders and mechanical or slope failures interact during dam collapses (Westoby et al., 2014a; Worni et al., 2014). The combined behavior of these processes, under heterogeneous natural conditions, makes it challenging to predict how a breach might develop,

and whether a complete collapse may occur in natural dams (Wahl, 2010; Walder and O'Connor,
1997), leading to modeling simplifications when trying to simulate such phenomena.
The resulting lake outburst floods, after breaching or overtopping of the moraine, comprise
highly unsteady flows that are characterized by pronounced changes as they propagate
downstream (Worni et al., 2012). To calculate downstream inundation caused by a GLOF event
requires the simulation of debris flow propagation, since sediment entrainment can cause the
volume and peak discharge to increase by as much as three times (Worni et al., 2014; Osti and
Egashira, 2009). Various 2D flow models have been used to simulate the downstream inundation
caused by a GLOF, including BASEMENT (Worni et al., 2012), which includes sediment
transport functions but no capabilities for debris flow simulation; FLO2D (Mergili et al., 2011;
Somos-Valenzuela et al., 2015) and RAMMS (Schneider et al., 2014), otherwise, do account for
debris flow.
As an interpretation of downstream consequences, flood hazard denotes potential levels of threat
as a function of intensity and likelihood of the arriving inundation (normally probability, but the
nature of avalanche events and other processes in the hazard chain restricts from assigning
numerical probabilities). Flood intensity is determined by the flow depth and velocity (García et
al., 2003; Servicio Nacional de Geología y Minería, 2007). Likelihood is inversely related to
magnitude, i.e., large events are less likely to occur (low frequency) than small events (Huggel et
al., 2004). Maps can be prepared that show the level of hazard resulting from the intensity of
various likelihood events. This allows communication of the flood hazard at various locations,
facilitating planning, regulation, and zoning based on the map while enhancing communication
to the affected community (O'Brien, 2012; USBR, 1988; FEMA, 2003).
This paper describes an analysis of the processes involved in a potential GLOF from Lake
Palcacocha in Peru and the resulting inundation downstream in the city of Huaraz. The simulated
process cascade starts from an avalanche falling into the lake resulting in a wave that overtops
the damming-moraine; the simulation continues with potential erosion due to moraine
overtopping and culminates with simulations of the ensuing downstream flooding and inundation
in Huaraz. In the following sections, the setting of the problem is presented, followed by
descriptions of the physical basis and modeling of each of the processes in the chain. The results
of each of the simulated processes are presented, concluding with details of the potential
inundation in Huaraz and hazard implications. Mitigation alternatives are investigated through an
analysis of several lake-lowering scenarios.

## 2   Study Area

Lake Palcacocha is located at 9°23′ S, 77°22′ W at an elevation of 4,562 m in the Department of
Ancash in Peru (Figure 1) and is part of the Quillcay watershed in the Cordillera Blanca. The
outlet of the lake flows into the Paria River, a tributary of the Quillcay River that passes through
the city of Huaraz. The Quillcay drains into the Santa River, the primary river of the region. The
lake had a maximum depth of 72 m in 2009 and an average water surface elevation of 4562 m
(UGRH, 2009).
The danger of a GLOF from Lake Palcacocha is paramount (HiMAP, 2014). A GLOF originating from
the lake occurred in 1941, flooding the downstream city of Huaraz, killing about 1800 people (according
to best estimates) (Wegner, 2014) and destroying infrastructure and agricultural land all the way to the
coast (Carey, 2010; Evans et al., 2009). The Waraq Commonwealth, a government body established
by the local municipalities of Huaraz and Independencia, was created to implement adaptation
projects related to climate change on water resources; at present, the Commonwealth is planning
a GLOF early warning system for Lake Palcacocha.
Prior to the 1941 GLOF, the lake had an estimated volume of 10 to 12 million $m^3$ of water
(Instituto Nacional de Defensa Civil, 2011). After the 1941 GLOF, the volume was reduced to
about 500,000 $m^3$ (Portocarrero, 2014). Lowering the level of glacial lakes is a common GLOF
mitigation practice in the Cordillera Blanca (Portocarrero, 2014). In 1974, drainage structures
were built at the lake to maintain 8 m of freeboard at the lake outlet, a level that at the time was
thought to be safe from additional avalanche generated waves. Nineteen years later, in March
2003, a landslide from the lateral moraine along the lake's southern side entered the lake,
launching a diagonal wave that traversed the lake and heavily eroded the reinforced dam. There
was a small outflow from the lake, but no serious damage occurred in Huaraz; however, the
event frightened the Huaraz city authorities. The regional government quickly repaired the
damaged structures (Portocarrero, 2014).
Lake Palcacocha continues to pose a threat, since in recent years it has grown to the point where
its volume is over 17.3 million $m^3$ (UGRH 2009). As shown in Rivas et al. (2015, Fig. 4), the
area of the lake has grown continuously from 0.16 $km^2$ in 2000 to 0.48 $km^2$ in 2012. Avalanches

from the steep surrounding slopes can reach the lake directly and potentially generate waves that could overtop and possibly erode the moraine dam, thus triggering a GLOF that could reach Huaraz (Hegglin and Huggel, 2008; Instituto Nacional de Defensa Civil, 2011). In 2010, Lake Palcacocha was declared to be in a state of emergency because its increasing water level was deemed unsafe (Diario la Republica, 2010; Instituto Nacional de Defensa Civil, 2011). Infrastructures at risk are spread between the lake and the city, including small houses, a primary school, fish farms, and water supply facilities. Siphons were installed in 2011 at the lake to temporarily lower the water surface of the lake by 3-5 m providing a total free board of about 12 m; however, further lowering of the lake to provide additional freeboard has been recommended (Portocarrero, 2014). Given the complexity of the problem and lack of information, local authorities and residents of Huaraz are concerned about the threat posed by the lake and have requested technical support to investigate the impacts that a GLOF could have on Huaraz and methods to reduce the risk. The latest hazard assessment for Lake Palcacocha (Emmer and Vilímek, 2014) has concluded that a GLOF resulting from moraine overtopping following an avalanche into the lake is likely; however, complete moraine failure resulting from an avalanche-generated wave is not likely, nor is moraine failure following a strong earthquake.

A recent 5 m *x* 5 m horizontal resolution Digital Elevation Model (DEM) of the Quillcay watershed generated by airborne LIDAR and new stereo aerial photographs was developed for this work by the Peruvian Ministry of Environment (Horizons, 2013) (Figure 1). Bathymetric data from a 2009 survey (UGRH, 2009) were combined with the surrounding DEM for the lake hydrodynamic and dynamic breach simulations.

## 3 Methodology

### 3.1 Overview

The methodology presented here considers a process chain similar to Worni et al. (2014) depicting an avalanche triggered GLOF from Lake Palcacocha to assess the potential inundation in Huaraz from such an event (Figure 2). The simulated avalanche originates from the Palcaraju glacier located directly above the lake. When an avalanche enters the lake, depending on its size and the level of the water surface in the lake, the resulting wave might overtop the damming-moraine and possibly initiate an erosive breaching process releasing considerable amounts of

water and debris into the Paria River and potentially inundating densely populated areas of
Huaraz downstream. The process chain from avalanche to inundation was simulated using four
models: potential avalanches were modeled using RAMMS (Christen et al., 2010), lake wave
dynamics were modeled with FLOW3D (Flow Science, 2012), the dynamic breaching process
was simulated in BASEMENT (Vetsch et al., 2006), and propagation of the flood wave
downstream and inundation in Huaraz were simulated in FLO2D (O'Brien, 2003).
The next sections describe each component for the framework used to simulate the
hazard process chain: avalanche simulation, wave simulation in the lake, moraine erosion
simulation, inundation simulation, and hazard identification.

## 3.2 Avalanche simulation

In non-forested areas, ice-rock avalanches can be generated on slopes of 30-50°,, and in tropical
areas the critical slope can be even less (Christen et al., 2005; Haeberli, 2013). Temperate
glaciers can produce ice avalanches if the slope of the glacier bed is 25° or more, but rare cases
with slopes less than 17° have occurred (Alean, 1985). The mountains surrounding Lake
Palcacocha have slopes up to 55°; therefore, they have a high chance of generating avalanche
events. Nonetheless, it is difficult to forecast when avalanches will occur and where the
detachment zone will be located (Evans and Clague, 1988; Haeberli et al., 2010).
The Rapid Mass Movements (RAMMS) avalanche model was used to simulate the progression
of avalanches down the mountain to the lake. RAMMS solves two-dimensional, depth-averaged
mass and momentum equations for granular flow on three-dimensional terrain using a finite
volume method (Christen et al., 2010; Bartelt et al., 2013). The inputs for the model include: (1)
terrain data (a DEM, described above); (2) fracture height; (3) the avalanche release area; and (4)
friction parameters. Descriptions of input parameters (2) - (4) and the criteria used to determine
their values are given in the following paragraphs. RAMMS computes the velocity of the
avalanche, the distance of the runout, the pressure distribution, and the height of the avalanche
front at different locations below the initiation point.
For the elevation of the Palcaraju glacier above Lake Palcacocha, the potential fracture type is
expected to be a slab failure or type I fracture as defined by Alean (1985). Huggel et al. (2004),
after Alean (1985), suggest that ice avalanches in slab failures are mainly produced in small and
steep glaciers with thicknesses between 30 to 60 m, where they are less frequent in large valley-
type glaciers. Alean (1985) shows examples of slab failure with thicknesses ranging from 19 to
35 m and volumes ranging from 1 to 11 million m³. The avalanche above Lake 513 that occurred
in 2010 is an example of this type of failure (Schneider et al., 2014). Following these precedents,
fracture heights of 25 m, 35 m and 45 m were selected for simulating the small, medium and
large avalanches respectively.
Three avalanche volumes are considered in this work, similar to the avalanche scenarios in
Schneider et al. (2014): $0.5 \times 10^6$ m³ (small), $1 \times 10^6$ m³ (medium) and $3 \times 10^6$ m³ (large). These
potential avalanche volumes are consistent with the elevations and slopes of the source area. The
release area (shown in Figure 3) was located at an elevation of 5200 m to the north east of the
lake following the main axis of the lake.
The friction parameters required by the RAMMS model are (1) the density of the rock and ice ($\rho$,
in kg m⁻³), (2) the Coulomb-friction term ($\mu$), and (3) the turbulent friction parameter ($\xi$) (Bartelt
et al., 2013). The Coulomb-friction term with a dry surface friction dominates the total friction
when the flow is relatively slow, and the turbulent friction parameter tends to dominate when the
flow is rapid, as is the case with the avalanches considered here (Bartelt et al., 2013; Christen et
al., 2010, 2008). The friction parameter values used in the RAMMS avalanche model are:
$\xi = 1000\,\text{ms}^{-2}, \mu = 0.12$ and $\rho = 1000$ kg m⁻³, values similar to those used to model the avalanche
into Lake 513 (Schneider et al., 2014).

## 3.3  Lake simulation

Impulse waves resulting from the impact of an avalanche with the lake were simulated with a
three-dimensional hydrodynamic model, FLOW3D (Flow Science, 2012). Much of the work in
impulse wave generation, propagation and run-up has been focused on empirical models that
replicate wave characteristics based on laboratory observations (Kamphuis and Bowering 1970;
Slingerland and Voight, 1979, 1982; Fritz et al., 2004; Heller and Hager, 2010). There have been
a few studies that perform numerical simulations of wave generation and propagation of slide-
generated waves, but most are still limited to simplified cases and two-dimensional simulations
employing the SWE (Rzadkiewicz et al., 1997; Biscarini, 2010; Cremonesi et al., 2011; Ghozlani
et al., 2013; Zweifel et al., 2006). However, the 2D SWE do a poor job of representing wave

generation and propagation because vertical accelerations cannot be neglected for slide-generated waves (Heinrich, 1992; Zweifel et al., 2006). Analytical calculations of wave runup and overtopping typically consider regular or simplified lake geometries (e.g., uniform water depth and constant slope of the terminal moraine) that do not necessarily hold true in natural reservoirs (Synolakis, 1987, 1991; Muller, 1995; Liu et al., 2005). Lake Palcacocha is very deep near the glacier with depths up to 72 m, but the last several hundred meters adjacent to the terminal moraine are very shallow with depths mostly less than 10 m (Figure 4). This discontinuous lakebed geometry significantly affects wave propagation and runup, making a hydrodynamic simulation necessary to represent the potential overtopping of the terminal moraine.

To overcome the limitations of analytical methods such as Heller and Hager (2010) in representing wave propagation, run-up and overtopping of the moraine, the three-dimensional hydrodynamic model FLOW3D (Flow Science 2012) was used to simulate the dynamics of avalanche-generated waves in Lake Palcacocha. The FLOW3D model grid used 400, 150, and 100 grid cells covering distances of 2400 m, 800 m, and 650 m in the x, y, and z directions, respectively. The RNG turbulence model with a dynamically computed mixing length and a fully three-dimensional, non-hydrostatic numerical scheme was used in the FLOW3D simulations.

The transfer of mass and momentum from the avalanche to the lake upon impact and the subsequent wave generation and propagation were simulated in FLOW3D by representing the avalanche as a volume of water equivalent to the avalanche volume that flows into the lake from the terrain above. Worni et al. (2014) and Fah (2005) approach the problem in the same way, simulating water instead of avalanche material. The density of the mixture of snow, rock and ice present in an avalanche is very close to the density of water (Schneider et al., 2014). Although the viscosities of the two fluids are different, this approximation of substituting water for the avalanche fluid is handled through adjustments in the model that compensate for any reduction in dissipation of energy due to the lower viscosity of water. To accomplish this, the results of the RAMMS avalanche model were used as calibration parameters; the depth of the avalanche fluid volume and height above the lake at which it is released were iteratively adjusted in FLOW3D until the velocities and depths of the avalanche fluid volume entering the lake matched the characteristics of the avalanche modeled in RAMMS. As long as the mass and momentum of the material hitting the lake in FLOW3D is similar to that of the RAMMS simulated avalanche, the

initial displacement wave should behave similarly as well; the water in the lake is pushed by the incoming avalanche, but the avalanche material does not reach the moraine, and the displaced wave is what propagates across the lake. Differences may arise for reflected waves since the avalanche material might settle in a different way over the lake's bed according to the avalanche properties (water representing avalanche material is more free to flow in the lake than actual rock-ice avalanche material). The primary output from the model is a hydrograph of wave overtopping discharge, if there is any, that is used as input to the downstream inundation model discussed later.

## 3.4  Moraine erosion simulation

Previous attempts at predicting outflow from potential failures of the Lake Palcacocha moraine have assumed, from a worst-case approach, that total or partial collapse of the moraine is possible (Somos-Valenzuela et al., 2014; Rivas et al., 2015). Although the history of GLOFs presents cases of large-scale breaches in diverse glacial settings, whether a total collapse at Lake Palcacocha is physically possible remains an unanswered question. To drain most of its impounded water, Lake Palcacocha requires a breach 985 m wide and 66 m deep, forming a continuous outlet at the front moraine (Rivas et al., 2015). Similar conditions resulted in moraine failure and subsequent outburst floods at Queen Bess Lake (Clague and Evans, 2000), Lake Ventisquero Negro (Worni et al., 2012), or Tam Pokhari Lake (Osti and Egashira, 2009). However, the morphology of Lake Palcacocha possesses a set of unique characteristics that could inhibit a large breach of its present moraine: (1) a reshaped morphology produced by the previous 1941 GLOF event and continuing glacier retreat, with a resulting irregular lake bed as an obstacle to flow; (2) a well-defined and curved outlet channel; and (3) a terminal moraine that resembles a long crested dam with an average width-to-height ratio of 14.9 (Rivas et al., 2015). Huggel et al. (2002, 2004) note that glacial lake damming-moraines with large width–height ratios (> 1.0) are much less vulnerable to overtopping and erosion by excess overflow or displacement waves. This paper seeks to go beyond previous work to determine the likelihood and potential magnitude of a moraine breach through hydro-morphodynamic simulations of the erosion process using BASEMENT.

Modeling the erosion of natural and artificial dams has been evolving since the early 1980's, when simple one-dimensional models based on empirical and parametric analyses were

developed to represent dam-breach processes, e.g., DAMBRK (Fread, 1988), WinDAM B
(Visser et al., 2011) and HR-BREACH (Hassan and Morris, 2012; Westoby et al., 2015). These
models describe breach phenomena by defining the rate of growth of a potential breach, then
including that breach definition in a hydrodynamic model (Rivas et al., 2015; Fread, 1984).
Although computationally efficient, one-dimensional models rely heavily on engineering
judgment and analysis of historical failure cases; when the expected breach shape, size and
growth rate are unknown, the models offer limited reliability to predict whether there will be
sufficient erosion to produce a breach at a particular site.
Erosion analysis in this paper evolves from the methods reported by Rivas et al. (2015), whose
performance evaluation of breach models focused exclusively on hydraulic considerations. That
partial perspective sets no physical limit on breach growth, assuming full moraine collapse is
possible (worst case scenario). This work, instead, applies a hydro-morphodynamic model to
describe the dynamic moraine erosion. Including this kind of analysis aims to explore the
possibility of full, partial or even null breaches according to flow characteristics but also
accounts for soil and morphological properties of moraines.
Many two-dimensional sediment transport models apply a SWE numerical scheme, in which
mobile bed meshes respond to shear stresses from hydrodynamic forces, and use empirical
functions of non-cohesive sediment transport that estimate drifting, entrainment, suspended
transport, bed load transport, and deposition of sediment. These models could potentially
simulate the moraine erosion process considered here. Models such as IBER (Bladé et al., 2014),
Delft3D (operated as a 2D model) (Deltares, 2014) and BASEMENT (Vetsch et al., 2014) follow
this scheme, and of these models, only BASEMENT is able to account for slope collapse as
erosion occurs and meshes change. BASEMENT has been implemented here to explore the
possibility of moraine erosion resulting from an overtopping wave.
Overtopping waves are potential triggers that might cause erosion of the terminal moraine at
Lake Palcacocha. Under wave transport conditions, vertical accelerations play an important role
in both water and sediment advection, influencing how overtopping waves might cause erosion
and possible failure of the moraine. Three-dimensional models can efficiently simulate flow
phenomena when those vertical accelerations are relevant. However, coupled erosion simulations
requiring additional hydro-morphodynamic functions possess additional challenges in three-

dimensional modeling. Several models combine three-dimensional numerical schemes (solving the Navier-Stokes equations) with sediment transport formulations, including Delft3D (Deltares, 2014), FLOW3D (Flow Science, 2014) and OpenFoam (Greenshields, 2015). FLOW3D (Wei et al., 2014) and OpenFoam (Liu and García, 2008) use the VoF (Volume of Fluid) technique to describe the solid-fluid interface, representing sediment beds as an additional fluid in multi-phase schemes. This approach seems successful for applications where the erodible bed remains submerged throughout the entire simulation or under steady flow conditions, but stability problems arise for cells exposed to drying and wetting periods (non-continuous submergence). Delft 3D avoids these stability issues by using a flexible mesh instead of a multi-phase approach to simulate changes due to erosion or deposition. The model, however, has limitations in representing fluid regions disconnected from boundaries (e.g., Lake Palcacocha at the center of the simulation domain, with no initial connection to the outlet downstream boundary and intermittent wetting and drying periods across the domain).

Worni et al. (2012) used BASEMENT to reproduce historic overtopping driven failures in Lake Ventisquero Negro. Despite the limitations of the BASEMENT two-dimensional SWE scheme, results show good agreement with the limited field data available, at least in terms of final breach dimensions. In this paper, BASEMENT was used for hydro-morphodynamic simulations of potential erosion-driven breach-failures at Lake Palcacocha. To overcome the two-dimensional SWE limitations of BASEMENT, results of three-dimensional hydrodynamic lake and overtopping wave simulations from FLOW3D were used as calibration parameters. The wave propagation and overtopping of the terminal moraine were simulated in both FLOW3D and BASEMENT. The zone of interest for BASEMENT simulations was at the terminal moraine, where erosion can occur and produce a moraine collapse. However, simulating the wave propagation across the whole lake moves the upstream boundary of the model, favoring a smoother transition at the interface between both models, where flow properties must match.

The BASEMENT model was started in the zone of the lake where wave generation occurs (wave splash zone in Figure 5), but the method of simulating wave generation was different from that used in FLOW3D because the flow characteristics at the inflow boundary must be artificially altered to compensate for the additional energy loss in the 2D shallow water equation (SWE) representation of BASEMENT. To facilitate comparison between the FLOW3D and BASEMENT models, hydrographs of results were compared at a common cross-section for both

models, located at the crest of the terminal moraine (target cross-section in Figure 5). Adjusting
the slope of the energy grade line at the upstream boundary (Figure 5) allowed an iterative
increase in momentum inflow until mass and momentum fluxes over the crest of the moraine
(target cross-section) matched the results from the FLOW3D simulations.
The relevant regions of the FLOW3D model, where fluid motion influences erosion and breach-
growth, are located near the moraine crest and downstream in the outlet channel. Through a
calibration procedure, the BASEMENT model was forced to replicate the hydrodynamic
conditions of the FLOW3D wave model. This was achieved by forcing momentum fluxes (that
are dissipated further downstream) at the inflow boundary of the BASEMENT model to be
unrealistically high. By adjusting energy slopes at the upstream boundary, momentum inflow
was iteratively increased until flow properties (mass and momentum fluxes) match the results
from full three-dimensional simulations according to hydrographs of discharge and velocity at
the crest of the artificial dam. This procedure aims to guarantee that BASEMENT can properly
model mass transport from wave phenomena despite the limitations of the two-dimensional SWE
simulations.
BASEMENT applies empirical functions to estimate erosion and deposition rates taking place
under the influence of flows from overtopping waves. Erosion resistance comes from soil
properties and the morphology of the bed. We have applied a hypothetical set of worst-case soil
conditions, intentionally decreasing the erosional strength in the Lake Palcacocha moraine. The
logic of this approach is that if breach simulations show no moraine collapse under the worst
possible conditions observed in the field, such collapse is unlikely to occur in real settings, where
the total soil matrix may contain soil that is more erosion resistant. This approach also seeks to
overcome a lack of independent erosion measurements, which makes any attempt at calibration
and further refinement of the breach model impossible.
The bed load transport is modeled with the single-grain Meyer-Peter and Müller (1948) (MPM)
model, which automatically discards any erosion resistance from hiding and armoring processes
occurring in multi-grain matrixes (Ashida and Michiue, 1971; Wu et al., 2000). Correction
factors to account for under- or over-prediction of the rate of bed load transport in the MPM
model range from 0.5 for low transport of sands and gravels to 1.7 for high transport cases
(Fernandez and Van Beek, 1976; Ribberink, 1998; Wong and Parker, 2006). A bed-load factor of
2.0 is used here, characterizing high sediment transport conditions. Table 1 displays the set of
sediment and slope failure characteristics used to build the Lake Palcacocha hydro-
morphodynamic model in BASEMENT. According to field data, coarser soils ($d_{50} \approx 19$ mm)
predominate at the walls of the outlet channel left by the 1941 GLOF at Lake Palcacocha
(Novotny and Klimes, 2014) where most of the outburst water would flow in a potential future
event. In agreement with the proposed hypothetical worst-case soil condition, a grain size of $d_{50}$
= 1 mm is assumed, representing characteristics of upper layer soils that may lead to significant
underestimation of erosion resistance.
**3.5   Inundation simulation**
One-dimensional models based on the St. Venant equations have been used to model the
downstream flood wave propagation of a GLOF. These models typically use the step-backwater
procedure, and cross-sectional averaged velocity and discharge (Westoby et al., 2014a).
Examples of this type of model include Klimes et al. (2013) who used HEC-RAS (USACE,
2010) to reproduce the 2010 GLOF from Lake 513 in Peru; Cenderelli and Wohl (2003) who
used HEC-RAS to reproduce steady-state aspects of GLOFs in the Khumbu region of Nepal;
Byers et al. (2013) who used HEC-RAS to model a potential GLOF from Lake 464 in the Hongu
valley of Nepal; Meon and Schwarz, (1993) who used DAMBRK (Fread, 1988) to model a
potential GLOF in the Arun valley of Nepal; and Bajracharya et al. (2007) who used FLDWAV
(NWS, 1998) to model a potential GLOF from Imja Lake in Nepal. Two-dimensional models
based on the depth-averaged SWE are often used to model downstream impacts of GLOFs
(Westoby et al., 2014a). Examples of applying this type of model include Worni et al. (2012)
who used BASEMENT to model flood propagation from a GLOF at Shako Cho Lake in India;
Schneider et al. (2014) who used RAMMS to model debris flow from an overtopping wave from
the 2010 GLOF event at Lake 513 in Peru; Somos-Valenzuela et al. (2015) who used FLO2D to
model downstream inundation from a potential GLOF from Imja Lake in Nepal; and Mergili et
al. (2011) used RAMMS to simulate debris flows and FLO2D to simulate floods and hyper-
concentrated flows from Lake Khavraz in Tajikistan.
FLO2D (FLO2D, 2012) is used to simulate the flooding downstream of Lake Palcacocha
considering debris flow that incorporates sediment characteristics (dynamic viscosity and yield
stress) as exponential functions of the sediment concentration by volume. Although the
simulation grid in FLO2D is two-dimensional, the flow is modeled in eight directions, solving
the one-dimensional continuity and momentum equations in each direction independently using a
central, finite difference method with an explicit time-stepping scheme. One of the advantages of
FLO2D is that for flows with high sediment concentration the total friction slope can be
expressed as a function of the sediment characteristics and the flow depth (FLO2D, 2012; Julien,
2010; O'Brien et al., 1993).
Due to the steepness of the terrain below Lake Palcacocha and low cohesion of the material from
the moraine, high velocities and turbulent flows with low dynamic viscosity and low yield stress
are expected (Julien and Leon, 2000). Therefore, from the empirical coefficients recommended
by FLO2D (2012) these two sets of parameters, that describe low yield stress and dynamic
viscosity respectively, are used: $\alpha_1 = 0.0765$, $\beta_1 = 16.9$, $\alpha_2 = 0.0648$ and $\beta_2 = 6.2$. Yield stress
and viscosity of the flow vary principally with sediment concentration based on empirical
relationships where the parameters $\alpha_i$ and $\beta_i$ have been defined by laboratory experiment
(FLO2D, 2012).
Downstream of Lake Palcacocha the flood will meet huge moraines in a steep canyon.
According to Huggel et al. (2004), erosion on the order of 750 $m^3$ $m^{-1}$ has been found in alpine
moraines. In the Andes and Himalaya, erosion cuts can be higher than 2000 $m^3$ $m^{-1}$, with peak
flow concentrations by volume on the order of 60-80%. Thus, given the uncertainties associated
to the calculation of the concentration of sediment, Huggel et al. (2004) recommend using an
upper limit for the average flow concentration by volume of 50-60%. This agrees with Schneider
et al. (2014), Julien and Leon (2000) and Rickenmann (1999) who recommend 50% sediment
concentration by volume, which is used in this study.
For the terrain elevation, a DEM was produced for this work (Horizons, 2013). Given the large
extent of the domain, running the inundation simulations on this 5 m x 5 m grid was impractical.
Therefore, the FLO2D simulations were run on a 20 m x 20 m grid.
Distributed roughness coefficient values were assigned based on land cover in the Paria basin
below Lake Palcacocha. Land cover was classified into five categories using the normalized
differential vegetation index (NDVI) from a multispectral image of a Landsat 7 image taken on
Oct 22, 2013 after reflectance correction and ISODATA analysis (Chander et al., 2009; Hossain
et al., 2009).
Given the lack of detailed information about the buildings and construction materials, an area
reduction factor of 20% was applied to account for the influence of buildings on the flow. Area
reduction factors are used in FLO2D to reduce the flood volume storage on grid elements due to
buildings or topography (FLO2D, 2012). Although FLO2D allows the inclusion of buildings and
obstacles that can affect the inundation trajectory, it was not clear in this work if the buildings of
Huaraz are strong enough to support the impact and, thus, deviate the flow. In some areas,
especially near the river, it is highly probable that the flow will destroy the buildings, but further
from the river that may be less likely to happen.
Flood intensity is determined by the resulting flow depth and velocity in Huaraz. Various
methods of determining the flood intensity from the flood depth and velocity have been
developed. The Austrian method (Fiebiger, 1997) uses the total energy of flow as the indicator of
intensity. The US Bureau of Reclamation (USBR, 1988) uses a combination of depth and
velocity and differentiates these for the impact on adults, cars, and houses. The Swiss method
(OFEE et al., 1997) defines intensity, independent of the object subjected to the hazard, as a
combination of depth and the product of depth and velocity.
In this work, the Swiss method is adopted to determine flood intensity as adapted for use in
Venezuela, where intensity thresholds were calibrated with field data from the 1999 alluvial
floods in Venezuela (PREVENE, 2001; García et al., 2002; García et al., 2005). Applying this
method requires simulating the different events to predict the spatially-distributed maximum
depths and velocities for each event, then transferring these results to GIS where a flood intensity
map for each event is created by applying the intensity categorization criteria, low, medium or
high (Table 2), to each grid cell in the map.
**3.6  Hazard identification**
Flood hazard is a function of intensity and likelihood of an event. In this case, the event is the
process chain resulting from an avalanche falling into Lake Palcacocha. The level of water in the
lake then determines the resulting wave that may or may not overtop the damming-moraine. To
determine flood hazard, normally probability would be the term used instead of likelihood, but
there is not enough data (i.e., recorded avalanche events) to assign probabilities to the different
avalanche events and other processes in the hazard chain; therefore, in keeping with other similar

studies (e.g. Huggel et al., 2004), a qualitative probability, or likelihood, is used. Likelihood is inversely related to avalanche magnitude; i.e., as discussed previously, large avalanches are less likely to occur than small avalanches. The flooding intensity for various likelihood events are used to prepare a preliminary hazard map that will allow communication to the affected community of the potential hazard at various locations and can facilitate planning, regulation, and zoning based on the map (O'Brien, 2012).

Following Schneider et al. (2014), Raetzo et al. (2002) and Hürlimann et al. (2006) the debris flow intensities have been classified into three classes, and an intensity-likelihood diagram was used to denote three preliminary hazard levels (Table 3). *High hazard* - people are at risk of injury both inside and outside buildings; a rapid destruction of buildings is possible. *Medium hazard* - people are at risk of injury outside buildings. Risk is considerably lower inside buildings. Damage to buildings should be expected, but not a rapid destruction. *Low hazard* - people are at slight risk of injury. Slight damage to buildings is possible. When multiple scenarios are considered, the highest hazard value for each cell is taken to create the preliminary hazard map (Raetzo et al., 2002).

## 4 Results

### 4.1 Avalanche simulation

The three avalanche events (large, medium and small) were simulated in RAMMS. The maximum heights of the avalanche material entering the lake range from 6 m for the small avalanche to 20 m for the large avalanche, and the maximum velocities range from 20 m s$^{-1}$ for the small avalanche to 50 m s$^{-1}$ for the large avalanche The RAMMS model simulation period was 60 seconds. The avalanches take from 33 to 39 seconds to reach the lake and the portion of the mass released that reaches the lake within the 60 second simulations ranges from 60 to 84% (Table 4).

### 4.2 Lake simulation

### 4.2.1 Current lake level scenario

For the three avalanche events listed in Table 2, FLOW3D simulations of the resulting wave generation, propagation and overtopping of the damming-moraine were run with the lake at the

current level of 4562 m. The wave simulations were analyzed for maximum wave height (measured in m above the initial lake surface) and compared to the wave heights calculated by the Heller and Hager (2010) method. Overtopping wave discharge hydrographs were calculated at the moraine crest mid-way between the artificial dam and the 1941 breach (Figure 3), and these hydrographs were used as calibration parameters for the dynamic breach model and as inputs to the downstream inundation model. The key results are summarized in Table 2 for each avalanche, including the overtopping volume, flow rate and wave height as the wave overtops the damming-moraine.

As the avalanche impacts the lake, it generates a wave that propagates lengthwise along the lake towards the damming-moraine and attains its maximum height when it reaches the shallow portion at the western end of the lake. The wave heights are shown in Table 4 for the height of the wave above the moraine crest at the point of overtopping and for the maximum mid-lake wave height. Although the mid-lake wave heights from FLOW3D are of the same order of magnitude as those calculated using the Heller and Hager (2010) method, the FLOW3D wave heights are all larger, with the difference in wave heights up to 13.3% for the large avalanche, and the difference is greater for small and medium avalanches. This may be an indication that the small and medium FLOW3D simulations overestimate the momentum transfer to the lake in the wave-generation process. However, the FLOW3D simulations are able to reproduce the avalanche characteristics of the RAMMs model as the avalanche enters the lake and account for lake bathymetry, likely giving more accurate results than the empirical method. In the FLOW3D results, the maximum wave height is attenuated approximately 30% before it reaches the damming-moraine. Normally, there would be a significant increase in wave height with the run-up against the terminal moraine, but because of the high dissipation of energy on the western end of the lake where it becomes shallow, this effect is somewhat lessened.

Looking in more detail at the wave propagation in the large avalanche scenario, there are two peaks in the wave height. The initial peak is about 1/3 of the way across the lake, corresponding to the empirical equations, and a higher peak occurs when the wave encounters the shallow portion of the lake. This is the beginning of the run-up process that culminates in the overtopping of the moraine, where the wave gains height as the water depth decreases.

The wave run-up causes a significant amount of water overtopping the damming-moraine. Figure
6 shows that the large avalanche results in an overtopping wave discharge hydrograph with a
peak of about 63,000 m$^3$ s$^{-1}$ approximately 60 s after the avalanche fluid is released and a smaller
peak of 6,000 m$^3$ s$^{-1}$ due to a reflected wave at about 200 s. The total overtopping volume was
1.8 x 10$^6$ m$^3$ for the large avalanche and 0.15 x 10$^6$ m$^3$ for the small avalanche (Table 4). The
duration of the initial wave of the avalanche events is about 100 seconds (large avalanche), 70
seconds (medium avalanche), and 50 seconds (small avalanche).
### 4.2.2  Lake mitigation scenarios
Two lake lowering or mitigation scenarios (with lake levels at 15 m and 30 m below the current
water level) were simulated to determine the impact on the moraine overtopping. Simulations for
all three avalanche sizes were repeated for each lake level and show that the overtopping wave
volume as well as the peak discharge of the wave are incrementally smaller as the lake is
lowered (Table 4). Although the overtopping volumes and peak flow rates decrease with
incremental lowering of the lake, the overtopping wave heights above the artificial dam increase.
This is due to several factors. First, as the point of avalanche impact is at a lower elevation with
lowered lake levels, there is more momentum in the avalanche fluid when it enters the lake.
Secondly, the stored volumes in the lake lowering scenarios are smaller, so the momentum
transfer to the lake per unit volume is higher, thus producing taller waves.
Although overtopping cannot be entirely prevented for the large avalanche events, even by
lowering the lake up to 30 m, the small avalanche shows no overtopping of the terminal moraine
for 30 m lake lowering, and the overtopping volume for the medium avalanche scenario is
reduced by 90% compared to the current level scenario. Overtopping is not avoided entirely for
the 15 m lake-lowering scenario; however, the overtopping flow rates and volumes are reduced
by about 60% and 80% for the medium and small avalanches, respectively, for 15 m lake
lowering.

## 4.3 Moraine erosion simulation

### 4.3.1 Hydrodynamic model

Dynamic simulations were made in BASEMENT using worst-case soil conditions described above (Table 1) and the large and medium avalanche wave dynamics to assess the erosion and potential breach of the damming-moraine at Lake Palcacocha. To validate the use of the two-dimensional BASEMENT model instead of the full three-dimensional FLOW3D model, the simulation results of the two models were compared using the peak differences between the mass and momentum fluxes and the normalized root mean squared error (NRMSE) (Table 2 - Table 5 in revised paper). The upstream boundary condition of the BASEMENT model was adjusted by varying inflow energy slopes to force the BASEMENT model to match the mass and momentum fluxes. Peak mass flux differences are low (ranging from 0.04% to 1.3%). Differences in peak momentum fluxes, however, show higher discrepancies. The NRMSE indexes assess the behavior of the entire hydrographs of mass and momentum fluxes and show a similar pattern to that of the peak fluxes, with errors between 2.0% and 3.8% for mass flux and 3.2% to 5.1% for momentum fluxes. Considering the extreme peaks of these simulations, the differences seem reasonable, making the corresponding BASEMENT models a good hydrodynamic base on which to build the erosion models (see next section). The relative agreement of the overtopping hydrographs between the BASEMENT and FLOW3D models shows that it is possible to replicate reasonably well the three-dimensional characteristics of avalanche-generated waves in a two-dimensional SWE model by exaggerating the energy slopes of upstream boundaries.

### 4.3.2 Hydro-morphodynamic model

Despite poor erosion resistance of the hypothetical soil matrix used in the simulations of the Lake Palcacocha damming-moraine, the results from the erosion simulations in BASEMENT with the lake at its current level indicate that a breach and total moraine collapse is extremely unlikely to occur. Both the large and medium avalanche events result in a no-breach development. Intense erosion takes place at the distal face of the moraine, where large-avalanche waves cause significant damage. The bed elevation of the outlet channel is lowered by up to 36 m at the distal face of the moraine; however, this vertical erosion does not propagate backwards toward the lake. Any significant erosion remains 270 m away from the lake surface with no

significant erosion and deposition areas occurring over the moraine crest (Rivas et al., 2015).
The apparent moraine stability seems to come from morphologic patterns the moraine geometry,
not from morphodynamic erosion resistance; the moraine does not fail in spite of its the very
erosive soil representing it in the hydro-morphodynamic model matrix. The peak flows at the toe
of the Lake Palcacocha damming-moraine (see Figure 3) have been attenuated to less than 50%
of the peak at the crest of the artificial dam.
The simulated scenario shows that a complete moraine failure with a large avalanche is
extremely unlikely, and any erosion that occurs as the wave passes the moraine does not
significantly affect the overtopping hydrographs. The large avalanche event is the worst case, so
if it doesn't fail then, it shouldn't fail for the medium and small avalanche events. The results
from the FLOW3D simulations were used as inputs to the downstream inundation model in
FLO2D.

## 4.4 Inundation simulation

Figure 1 shows the locations of 5 cross-sections downstream of Lake Palcacocha where
hydrographs are reported from the FLO2D simulations. Figure 7 and Table 6 show the results of
the simulation of the large avalanche with the current lake level. At cross-section 1, the
hydrograph is still similar to the original hydrograph at the lake with a high-intensity peak flow
that is of relatively short duration. The flow is quickly attenuated as it moves downstream, and
the hydrograph at cross-section 2, located just upstream of the point where the river canyon
narrows and becomes steeper, has a much lower peak than the overtopping hydrograph at the
lake, but it is of longer duration. This is expected because the river is relatively wide with gentle
slopes between the lake and cross-section 2.
Cross-section 4 is located at the entrance to the city of Huaraz. The peak discharge of the large
avalanche event diminishes about 40% between the cross-sections 2 and 4. From the beginning
of the large avalanche event it takes the flood wave about 1.3 h to reach cross-section 4 (Table
6), and the peak flow arrives shortly after. The peak flow takes about 0.75 h to cross the city to
cross section 5 and the peak is attenuated by about 50% in the crossing. Values for the medium
and small avalanche events are shown in Table 6. They take considerably longer to arrive and
cross the city, but their peaks are attenuated about 50% as well. The resulting maximum flood

intensities in Huaraz are shown in Figure 8 for the current lake level and two lake mitigation scenarios (15 m and 30 m of lake lowering) and each of the three avalanche scenarios. The highest intensity areas are near the existing channels of the Quillcay River and the Rio Santa on the south side of the river.

## 4.5 Hazard identification

Preliminary hazard identification uses the flood intensity maps (Figure 8) and converts them to maps showing the hazard level at different points in the city according to the intensity-likelihood flood hazard matrix shown in Table 3. The resulting hazard is obtained by combining the three avalanche events into a single preliminary hazard map selecting the highest hazard for each cell, which reflects the result of all the possible avalanche combinations (Figure 9).

## 4.6 Probable maximum inundation

The BASEMENT modeling results (see Sect. 4.3.2. hydro-morphodynamic model) indicate that the overtopping wave generated from the large avalanche event does not cause sufficient erosion to initiate a breach of the moraine and release the lake water, thus rendering a full collapse of the moraine extremely unlikely. The authors consider this scenario nearly impossible given the current understanding of the moraine conditions and the extensive modeling of the moraine using extremely erosive soil characteristics. The decision which scenario to eventually include in a hazard map is not just a scientific question, but also a political one. The results of the breaching scenario are included since they are needed in order to assess the worst-case scenario, something science and engineering must communicate to the decision makers and stakeholders. For the sake of providing complete information, the probable maximum flood as a result of a full breach of the damming-moraine at Lake Palcacocha was simulated, assuming this event is the worst possible scenario that could conceivably occur. This probable maximum flood is estimated by modeling the event of a full collapse of the moraine following an overtopping wave generated by a large avalanche that erodes the moraine to the extent that the release of the lake water can maintain the erosion and create a full breach of the moraine. The HEC-RAS breaching model (USACE, 2010) was used to simulate the progression of the breaching process and the resulting breaching hydrograph (Rivas et al., 2015). The inflow hydrograph for downstream simulations of

this scenario was created by combining the large avalanche overtopping wave hydrograph under
current lake level conditions with the HEC-RAS breach hydrograph.
The flood intensity resulting from this scenario is illustrated in Figure 10. The flood
hazard is not computed since the likelihood of the medium and small avalanches generating
waves capable of eroding the moraine to the extent of initiating a breaching process are simply
too remote to consider.

## 4.7   Sensitivity analysis

A sensitivity analysis of the inundation was performed, and it focused on three components: (1)
sediment concentration by volume, (2) rheology of the flow, and (3) roughness.
**Sediment concentration:** The sediment concentration is an important factor in simulating the
inundation in Huaraz because it affects the volume of the flow, and consequently the depth of
inundation (Somos-Valenzuela, 2014). A potential GLOF will erode the bank along the river,
especially where lateral moraines are present (cross-section 3), scouring, transporting and
depositing soil many times as the flood moves downstream from the lake to the city. FLO2D
does not represent this process when using the Mudflow module. Additionally, we did not have
field information to perform a study of these effects. Therefore, in this work, a fixed sediment
concentration of 50% by volume was used, which is a good upper limit according to the
literature and the FLO2D developers (FLO2D, 2012), but it may be too high if the material
available for erosion is not sufficient in the inundation path. Analysis of sensitivity to sediment
concentration was performed for the inundation in Huaraz, assessing the affect on velocity and
flood stage with sediment concentrations of 0, 20, 30, 40 and 50% (Somos-Valenzuela, 2014).
The flood wave travel times were similar for all cases, and the depths increased with sediment
concentration due to the increased volumes (an increase of up to 8 m at cross-section 4 for a
concentration of 50% compared to no sediment). Thus 50% concentration was considered a
reasonable value to use, and it gives a conservative result.
**Flow rheology:** With regard to the possible effects and limitations in the model settings
associated with different flow rheologies, we identified two major sources of uncertainty: (1) the
physical characteristics of the mixture and (2) the volume of material that will be eroded,
transported and deposited again, a process that may happen many times during the trajectory of

the flood. FLO2D can simulate the behavior of the mixture assuming that it won't change throughout the simulation. Consequently, it is not able to consider transformations of the flow rheology; however, changes in concentration by volume can change the dynamic viscosity ($\eta$) and yield stress ($\tau_y$) (O'Brien and Julien, 1988). Additionally, scouring is not simulated in the FLO2D mudflow module, so we prescribe the concentration by volume to be 50% based on the literature recommendations.

The quadratic rheological model used within FLO2D combines four stress components of hyper-concentrated sediment mixtures: (1) cohesion between particles; (2) internal friction between fluid and sediment particles; (3) turbulence; and (4) inertial impact between particles, where the cohesion between particles is the only parameter that is independent of the mixture concentration or hydraulic characteristics (Julien, 2010:243; O'Brien and Julien, 1988). According to the few studies of the composition of the Lake Palcacocha moraine (Novotný and Klimeš, 2014), the cohesion can be considered nearly equal to zero, which implies that the resulting mixture would have low yield stress and dynamic viscosity. Consequently, from the list of 10 soils presented in the FLO2D manual (FLO2D, 2012: Table 8, p. 57), we selected parameters that give a low yield stress and dynamic viscosity (Glenwood 2). In addition, a sensitivity analysis was performed using the parameters for the other soils listed in Table 1 (Aspen Pit 2, Glenwood 1, and Glenwood 3 with higher dynamic viscosities and yield stresses, and Glenwood 4 with much higher values). The results of the sensitivity analysis (FLO2D simulations) show that the flood arrival time at cross section 4 varies from 1.05 to 1.32 hours (compared to 1.32 hours with Glenwood 2 parameters, see Table 6 in original paper). The peak flow varies from 1954 to 3762 $m^2$ $s^{-1}$ (compared to 1,980 $m^3$ $s^{-1}$ using Glenwood 2). The Glenwood 4 parameters result in the shorter arrival time (somewhat counter-intuitively) and higher peak value. Therefore, the rheology, which is a function of the concentration of the mixture and the soil characteristics, does affect the travel time and the peak flows. The results are not expected to be highly sensitive if the dynamic viscosity were to be lower than what was assumed (Glenwood 2), which is expected from the few soil studies in the area.

The model results show that the flood takes about 45 minutes to cross the city (travel of front of inundation between cross-section 4 and 5) and the peak flow takes 55 minutes to cross the city. The inundation spreads through the city diffusing the peak flow and reducing it considerably.

Sensitivity analysis showed that increasing the dynamic viscosity, from Glenwood 2 to
Glenwood 4, the flow travels faster, arriving at the city 17 minutes earlier, crossing the city in 36
minutes, with the peak flow taking 45 minutes to cross the city. Glenwood 2 and 4 are the lower
and higher end, respectively, for the dynamic viscosity parameters used in the sensitivity
analysis.
**Roughness:** The impact of roughness was analyzed in the dissertation of Somos-Valenzuela
(2014) who concluded that travel time is sensitive to roughness, increasing by 1.5 hours for
travel from the lake to cross-section 4 if the roughness is increased from 0.1 to 0.4. Also, the
peak flow is inversely proportional to the roughness, so lower roughness results in a slightly
higher peak (less than 10% difference in peak flow for 0.1 vs. 0.2 roughness coefficients)
(Somos-Valenzuela, 2014). When the roughness within the city is reduced to 0.02, the minimum
value recommended for asphalt or concrete (0.02-0.05) (FLO2D, 2012) and the 20% area
reduction factor is removed (so the flood is limited only by the topography), the inundation takes
22 minutes to cross the city, 50% of the originally computed time. This is an unrealistic value
since it considers the entire land cover of the city to be asphalt with no disturbances, buildings,
streets, trees, debris, etc.; however, this can be considered a minimum possible time for the flood
to cross the city. If a roughness value of 0.05 is used, then the inundation takes 26 minutes to
cross the city, and if 0.1 is used, a low but more realistic value, the flood takes 36 minutes to
cross the city. Thus, the travel time across the city is more sensitive to changes in roughness
values than rheology characteristics.
The relative impacts of the GLOF process components can be seen by analyzing the inundation
in the city of Huaraz for each of the scenarios simulated. The avalanche size may have the most
significant impact on downstream flood hazard. With the lake at its current level, the affected
area in Huaraz for the small avalanche scenario ($0.7 \text{ km}^2$) is approximately 35% of the area
potentially affected by the large avalanche ($2.0 \text{ km}^2$). The other process that could significantly
influence the flood hazard in the city is the erosion of the damming-moraine. Although results
from this work indicate that a complete moraine failure is extremely unlikely, the possibility of a
catastrophic breach cannot be categorically excluded based on existing evidence. If such a breach
were to occur, the inundated area could increase to $4.93 \text{ km}^2$, almost 246% more than the large
avalanche–no breach scenario ($2 \text{ km}^2$). Considering the results of the lake lowering mitigation
scenarios, the reduction in hazard area in Huaraz is mostly in the high hazard zones (see Table
7). There is a 27% and 45% reduction in the high hazard area (compared to the current lake
level) when the lake is lowered 15 or 30 m, respectively.
**5    Discussion**
**5.1    General discussion**
In this paper, each step in the hazard process chain that could lead to inundation of Huaraz from
a GLOF from Lake Palcacocha has been simulated. Of the simulation methods used in this work,
the lake hydrodynamics and moraine erosion models are advancements beyond what has been
previously reported for GLOF hazard process chain simulations. The use of a fully three-
dimensional hydrodynamic model for simulating wave generation, propagation, run-up and
overtopping of the damming-moraine allows predictive modeling of the process chain through
better representation of the physical processes. Other studies (e.g., Schneider et al., 2014) have
used a past event to calibrate the models and then used those calibrations for predictive modeling
of other scenarios. When data for past events are not available, the three-dimensional model can
help overcome the limitations of two-dimensional SWE models. Better representation of the
physical processes in the model (i.e., three-dimensional non-hydrostatic) makes the models
useful for predictive purposes without a heavy reliance on calibration. Modeling for predictive
purposes, such as that presented in this paper, are useful for analyzing potential GLOF impacts
and mitigation strategies.
The general lack of field data regarding actual GLOF events leads to many unknowns about the
processes, particularly processes related to avalanches, lake dynamics and moraine erosion.
Previous simulations of GLOFs have focused on calibrating upper-watershed processes based on
post-event observations (Schneider et al., 2014), but there is very little information on avalanche
characteristics, magnitude of avalanche-generated waves (Kafle et al., 2016), or erosive
capabilities of overtopping waves on which to base validation of these simulated processes.
There is still a considerable amount of uncertainty in the 3D modeling approach for avalanche-
generated waves. Nonetheless, even post-event field studies of GLOF waves have difficulty
accurately characterizing the wave magnitudes. The 3D modeling approach presented in this
paper is intended as an alternative to partially overcome the absence of field data from a GLOF
event at the location of the study.

## 5.2  Model calibration

Because field data are not available, we attempted to counteract the inability to calibrate the models by using the best available physical representations in our modeling approach. The 3D hydrodynamic model and the hydromorphodynamic model of moraine erosion can give us a better understanding of the likely outcomes of these processes than models that require extensive calibration (e.g., 2D SWE models and breach simulations such as reported in Rivas, et al. (2015)). This is not to say that these models are free from significant uncertainties, but as a model provides better mechanisms to represent the underlying physical phenomena, uncertainties move from the model engine to the physical initial and boundary parameters, reducing the amount of physical or empirical assumptions. Caution is required in any case because lacking a means of calibration/validation, these results represent estimations that might deviate from reality without proper analysis or judgment.

Simulations of lake dynamics with a three-dimensional non-hydrostatic model (FLOW3D) and a two-dimensional SWE model (BASEMENT) indicate that the SWE approximation is not adequate to simulate waves generated by avalanches because of the large energy dissipation due to significant vertical accelerations. Two-dimensional hydrostatic models may be adequate for simulating past events where calibration parameters based on field data may be used to overcome the approximations in the SWE model (Schneider et al., 2014), but it is important that calibration be performed at appropriate points in the model to account for energy dissipation as the wave propagates across the lake. The results from the BASEMENT simulations suggest that, without careful setting and adjustment of the model's boundary conditions, two-dimensional models might produce unrealistic results for wave driven phenomena that underestimate the magnitude of an event. Reference simulations, like those from three-dimensional hydrodynamic models, may help to overcome limitations on the two-dimensional models and turn them into more flexible and efficient tools for erosion and breach failure assessment.

The primary limitation of the lake hydrodynamic model arises from representing an avalanche entering the lake as a volume of water, rather than a combination of rock, ice and snow (Kafle et al., 2016). The wave model calibration method involves controlling the height and depth of the release area in order to influence the fluid height and velocity in the model as the avalanche enters the lake. This helps to overcome the limitations of substituting water for the avalanche

fluid mixture, but the water representation does not dissipate the energy in the same way as the true avalanche mixture, and the mixing of the avalanche fluid with the lake is not accurately represented in the model.

The lake model has a considerable amount of uncertainty. The greatest sources of uncertainty are the avalanche characteristics (inputs to the lake model) and the wave generation. The processes associated with wave generation from avalanche impact are poorly understood, and current model limitations do not allow for an avalanche to be simulated with its actual flow characteristics (rheology, density, etc.) in the same environment as the lake dynamics. Therefore, it is difficult to represent wave generation in a fully physical manner. The avalanche characteristics (depth and velocity) have a significant impact on the wave characteristics and moraine overtopping hydrograph. Additionally, the method of representing the avalanche impact boundary condition may overestimate the momentum of the inflow; the result of this may be somewhat larger wave height, but the greatest impact is in the peak flow and total volume of the overtopping wave. The highest estimates of the overtopping wave characteristics are presented in the paper to illustrate a worst-case scenario, but it is likely that the actual magnitude of an avalanche generated wave may be less than what is reported here.

## 5.3 Worst-case event simulation

The moraine erosion simulations used a worst-case approach, depicting the moraine as a structure with very low erosive resistance. Therefore, the resulting moraine erosion is overestimated, i.e., erosion depth, width, length, and growth rate. Thus, the simulations sacrifice accuracy in modeling the erosion process to gain confidence in predicting the potential for moraine breaching and collapse. The erosion simulation results suggest that the Lake Palcacocha damming-moraine has adequate stability to resist erosion induced by large waves, since the modeled erosion does not reach from the distal face back to the lake, which would allow the lake water to flow through the breach and accentuate the erosion process and lead to possible moraine failure. The main source of erosive resistance in the simulations is from the morphology of the moraine (e.g., large width to height ratio, long crested dam, and gentle slope of distal moraine face) and not from soil resistance. Previous qualitative assessments of the Lake Palcacocha moraine (Emmer and Vilímek, 2013) and similar structures at other lakes (Worni et al., 2014) assigned very low probabilities of failure of the moraine, but did note its high susceptibility to

wave overtopping. This study, however, provides the first quantitative assessment of possible breach failure for the damming-moraine at Lake Palcacocha, reinforcing results from the qualitative assessments by using numerical simulations that account for the morphology of both the lake and moraine in a two-dimensional modeling scheme.

The functions in BASEMENT to simulate erosion come from empirical equations of sediment transport developed for fluvial environments. Due to their empirical nature, the equations depend on calibration to achieve accurate results of erosion and deposition rates. Worni et al. (2012) showed that BASEMENT can achieve realistic results using soil parameters that resemble actual moraine properties. The bed-load transport model used in this paper (Meyer-Peter and Müller, 1948) has been derived in different forms since its first release to reverse the model's tendency of over predicting erosion. Newer bed-load models address this problem by applying a direct reduction factor on resulting transport rates or adding hiding functions to account for multi-grain soil matrixes (e.g. Ashida and Michiue, 1971; Wu et al., 2000). Additionally, the two-dimensional limitation of BASEMENT restricts its application for problems where vertical accelerations are relevant, or vertical flow distribution is not uniform. Under these latter conditions, BASEMENT needs three-dimensional simulations to serve as calibration parameters before applying the model to predict erosion and breach formation.

Even though a prescribed terminal moraine collapse scenario was simulated, it was not included in the preliminary hazard map for two reasons. First, the complete collapse scenario is based on the premise that we should consider a worst case scenario, but we could not initiate the moraine collapse using our numerical approach; even when a large overtopping wave and highly erosive materials were assumed, the width of the moraine is simply too great, and the erosion does not extend from the distal face of the moraine back to the lake. Therefore, we artificially prescribed and simulated the moraine collapse. Using empirical equations we determined the time that the collapse will take and the hydrograph was calculated following hydrodynamic constraints as indicated in Rivas *et al.* (2015). Based on these modeling results it is extremely unlikely that the collapse will occur, but it cannot be completely disregarded. Secondly, given the magnitude of the extremely unlikely breach scenario results, it is important to avoid creating confusion as a result of misinterpretation of the results. People in Huaraz should decide if they want to consider the worst case scenario in their planning, and this work is limited to informing that decision making process.

## 5.4 Comparison to 1941 GLOF

There are still many unknowns about the 1941 event, including the precise lake volume at that time, underlying bathymetry and pre-GLOF moraine morphology, flood volume and discharge hydrograph; aerial images and derived historical maps represent the only sources of information, known to the authors, about the pysical characteristics of the 1941 GLOF, providing at least a rough visual estimation of the flood area. In a qualitative comparison with the GLOF from 1941, we used a map published by the Instituto Nacional de Defensa Civil (INDECI, 2003) where three mudflow event extensions are delineated: Aluvion Preincaico, Aluvion Huallac and Aluvion Cojup 1941. In Figure 11 we plot the inundation extension reported in this paper on the map of the 1941 event delineated by INDECI (2003) and confirm that the inundation modeled has reasonable dimensions in comparison with this historical information. The volume at the time was estimated to be on the order of 14 million $m^3$ (Vilímek et al., 2005), which is more than 7 times the volume that we have calculated for the large avalanche (1.8 million $m^3$). This may explain the fact that in our results the inundation does not pass out of the bank from the Cojup River to the Quilcaihuanca River in the area where the rivers are very close together near the entrance to the eastern border of the city. However, these results demand caution; a qualitative comparison only describes potential differences between simulated and observed flood areas. Because the moraine failure in 1941 changed the upstream conditions at Lake Palcacocha, historical aerial images of flooded areas constitute no source of information for precise calibration for our model.

## 5.5 Lateral moraine collapse in 2003

According to Vilímek *et al.* (2005), the lateral moraine collapse that occurred in 2003 at Lake Palcacocha was due to a wave produced by a landslide on the internal face of the left lateral moraine that was triggered by extensive rainfall precipitation which over-saturated the moraine material. The terminal moraine was eroded but it did not breach. A downstream flood was produced by the water that overtopped the moraine. While this type of landslide from the lateral moraine is likely to occur again in the future, the work reported here focuses on the potential effects of an avalanche-generated wave because the magnitude of landslides likely to enter the lake are less than the avalanche volumes we have considered, and the effect of a landslide-generated wave may be somewhat mitigated as it propagates diagonally across the lake, whereas

an avalanche-generated wave would enter along the longitudinal axis of the lake and is unlikely
to be attenuated by reflections off of the lateral moraines.

## 6   Conclusions

There is consensus among local authorities, scientists and specialists that Lake Palcacocha
represents a GLOF hazard with potentially high destructive impact on Huaraz, and this
consensus has been validated by the modeling results presented in this paper. Huaraz previously
experienced a GLOF in 1941 when the outburst from Lake Palcacocha killed about 1800 people
(Wegner, 2014). However, there was no previous model that assessed the potential extent of
inundation given the current size of the lake. This work used high-resolution topographic
information in a two-dimensional debris flow model of the inundation below the lake. Several
avalanche magnitudes were used to assess the range of possible inundation and hazard in Huaraz.
In addition, scenarios of based on lake lowering were simulated to determine the mitigation
potential of lowering the lake level.
This work has provided a physical analysis of all of the processes in a chain of events from the
summit to the city for a potential GLOF from Lake Palcacocha and determined that there could
be significant impacts in the city of Huaraz. This work has demonstrated advancements in
simulation methods for the lake dynamics and the dynamic erosion process of the damming-
moraine that help further our understanding of this type of event. Based on the results of this
work, it can be concluded that three-dimensional non-hydrostatic simulations of slide-generated
waves are necessary to capture the full effects of these waves and their magnitudes at the point of
overtopping. This study also found that the morphology of the damming-moraine at Lake
Palcacocha may be a more important factor than the soil erosion characteristics in determining
the stability of the moraine and its ability to withstand the high forces of large overtopping
waves.
The results indicate that a GLOF for a large avalanche event takes about one hour and twenty
minutes to arrive at the city (cross-section 4) after the avalanche process starts, and the flood
peak arrives two to three minutes later. The peak crosses the city from in about 45 minutes,
expanding to the north and south as it progresses through the city. Based on the flood intensity,
the most highly impacted areas in the city are near the Quillcay River just to the south of the

river. While the inundated areas for medium and small avalanches are less than the affected area due to a large avalanche, there is a significant reduction in the high intensity areas for these events. For the large avalanche event, most of the affected area of the city has a very high hazard level for the current lake level. With mitigation through lake lowering, the total affected area is reduced (by around 30% for a 30 m lowering scenario), but the greatest impact of lake lowering is that more of the high and medium hazard zones areas are downgraded to low hazard. The results indicate that Lake Palcacocha is dangerous if an avalanche occurs, especially since there is no way to prevent an avalanche from falling into the lake and overtopping waves are expected for all avalanche sizes with the lake at its current level. The damage could be even more extensive in the extremely unlikely event of an avalanche and moraine breach.

Based on these conclusions, it is recommended an early warning system should be installed in the basin. This is an urgent matter because a significant area of the city of Huaraz could be impacted by a GLOF from Lake Palcacocha, and timely warning and evacuation of the population is the best way to prevent injuries and mortalities. The results of this study indicate that the inundated area may be reduced through lake lowering, and the highest likelihood event (small avalanche) produce significantly less inundation with lake lowering. An economic analysis of mitigation alternatives should be undertaken to determine an optimized lake level that balances cost and potential benefits.

## ACKNOWLEDGEMENTS

The authors acknowledge the support of the USAID Climate Change Resilient Development (CCRD) project and the Fulbright Foundation for the support of Somos-Valenzuela and Rivas. The support of the software developers from FLO2D Software, Inc., Flow Science, Inc., and RAMMS made much of the work reported here possible. The support of Josefa Rojas and Ricardo Ramirez Villanueva of the IMACC project of the Peruvian Ministry of Environment provided valuable assistance in obtaining the new DEM of the Quillcay watershed. Prof. Wilfred Haeberli, Dr. Alton Byers and Dr. Jorge Recharte provided valuable insights and encouragement through the entire work. Likewise, we highly appreciate readings and feedback on the sections of dynamic breach simulations from Adam Emmer. The authors greatly appreciate the constructive comments of Dr. Christian Huggel and one other anonymous reviewer.

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

1    Table 1. Main Parameters Defining the Soil Matrix Used in BASEMENT Simulations of the

2    Lake Palcacocha Moraine.

| Morphodynamic parameter | Adopted value | Source |
|---|---|---|
| Sediment transport formula | MPM single-grain | Meyer-Peter and Müller (1948) |
| Diameter $d_{50}$ | 1 mm | Novotny and Klimes (2014) |
| Porosity | 40% | Typical value for spherical sediment |
| Bed load factor | 2 | Modified from Wong and Parker (2006) and Worni et al. (2012) |
| Failure angle of submerged sediment | 36.5 degrees | Novotny and Klimes (2014) |
| Failure angle of dry sediment | 77 degrees | Worni et al. (2014) |
| Failure angle of deposited sediment | 15 degrees | Worni et al. (2014) |

4    Table 2. Flood Intensity Classification.

| Intensity | | Maximum Velocity (m s$^{-1}$) times Maximum Depth (m) | | | | Flood Intensity |
|---|---|---|---|---|---|---|
| | | > 1.0 | 0.2 - 1.0 | < 0.2 | | |
| Maximum Depth (m) | > 1.0 | High | High | High | | High |
| | 0.2 - 1.0 | High | Medium | Low | | Medium |
| | < 0.2 | High | Low | Low | | Low |

Table 3. Flood Hazard Classification.

| Hazard | | Likelihood | | |
|---|---|---|---|---|
| | | High | Medium | Low |
| | | Avalanche Size | | |
| | | Small | Medium | Large |
| Intensity | High | High | High | High |
| | Medium | High | Medium | Low |
| | Low | Medium | Low | Low |

| Hazard Level |
|---|
| High |
| Medium |
| Low |

Table 4. Characteristics of Three Avalanche Events of Different Size as Simulated in RAMMS.
Overtopping Volume, Flow Rate and Wave Height for Three Avalanche Events as Simulated in
FLOW3D for the Current Lake Level and Three Lake Mitigation Scenarios. Comparison of mid-
lake wave heights between Heller and Hager (2010) equations and FLOW3D simulations for 0-
m lower scenario

| | Avalanche Event | | |
|---|---|---|---|
| | Large | Medium | Small |
| **Avalanche characteristics in RAMMS** | | | |
| Avalanche size ($10^6$ m$^3$) | 3 | 1 | 0.5 |
| Maximum depth of avalanche material at lake entry (m) | 20 | 15 | 6 |
| Maximum velocity of avalanche material at lake entry (m s$^{-1}$) | 50 | 32 | 20 |
| Time to reach the lake (seconds) | 33 | 36 | 39 |
| % of mass released that reaches the lake in 60 seconds | 84 | 72 | 60 |
| **0 m lower** | | | |
| Overtopping volume ($10^6$ m$^3$) | 1.8 | 0.50 | 0.15 |
| Overtopping peak flow rate (m$^3$ s$^{-1}$) | 63,400 | 17,100 | 6,410 |
| Overtopping wave height above artificial dam (m) | 21.7 | 12.0 | 7.1 |
| Maximum mid-lake wave height (m) - Heller and Hager (2010) | 42.2 | 21.1 | 8.8 |
| Maximum mid-lake wave height (m) – FLOW3D | 47.8 | 30.1 | 19.6 |
| **15 m lower** | | | |
| Overtopping volume ($10^6$ m$^3$) | 1.6 | 0.2 | 0.02 |
| Overtopping peak flow rate (m$^3$ s$^{-1}$) | 60,200 | 6,370 | 1,080 |
| Overtopping wave height above artificial dam (m) | 38.4 | 27.5 | 25.1 |
| **30 m lower** | | | |
| Overtopping volume (m$^3$) | 1.3 | 0.05 | 0 |
| Overtopping peak flow rate (m$^3$ s$^{-1}$) | 48,500 | 1,840 | 0 |
| Overtopping wave height above artificial dam (m) | 60.8 | 42.5 | 0 |

Table 5 . Fit indices for flow properties at the overtopping zone of Lake Palcacocha (Target cross
section in Figure 5) comparing BASEMENT and FLOW3D simulation results.

| Flow property | Fit indices | Scenarios | |
|---|---|---|---|
| | | No lake lowering | Lake lowering |
| Mass flux | Peak mass flux difference (%)* | 0.04 | 1.3 |
| | NRMSE (%)** | 3.8 | 2.0 |
| Momentum flux | Peak momentum flux difference (%)* | 7.3 | 4.4 |
| | NRMSE (%)** | 5.1 | 3.2 |

* Peak differences refer to relative errors (expressed as percentage) between point measurements of
maximum mass flux and momentum flux for both models (Flow 3D and BASEMENT).
** NRMSE = Normalized Root Mean Square Error, accounts for errors across the entire hydrograph of
mass and momentum fluxes.
Table 6. FLO2D Simulation Results at Cross-sections Downstream of Lake Palcacocha for the
Current Lake Level and a Large Avalanche.

| Cross Section | Avalanche size | Arrival time (hr) | Peak time (hr) | Peak discharge ($m^3 s^{-1}$) |
|---|---|---|---|---|
| **1** | Large | 0.05 | 0.05 | 39,349 |
| | Medium | 0.08 | 0.09 | 4,820 |
| | Small | 0.14 | 0.16 | 436 |
| **2** | Large | 0.51 | 0.65 | 3,246 |
| | Medium | 1.07 | 1.14 | 347 |
| | Small | 2.8 | 2.88 | 27 |
| **3** | Large | 0.81 | 0.84 | 2,989 |
| | Medium | 1.67 | 1.71 | 272 |
| | Small | 4.57 | 4.6 | 19 |
| **4** | Large | 1.32 | 1.36 | 1,980 |
| | Medium | 2.9 | 2.97 | 149 |
| | Small | 8.68 | 8.73 | 8 |
| **5** | Large | 2.1 | 2.26 | 920 |
| | Medium | 4.95 | 5.27 | 73 |
| | Small | 15.8 | 16.1 | 4 |

Table 7. Areas of Each Hazard Level corresponding to the Current Lake Level and Two Lake
Mitigation Scenarios.

| Mitigation | Low hazard area ($km^2$) | Med. hazard area ($km^2$) | High hazard area ($km^2$) | Total affected area ($km^2$) |
|---|---|---|---|---|
| 0 m lower | 0.52 | 0.05 | 1.43 | 2.01 |
| 15 m lower | 0.61 | 0.00 | 1.04 | 1.65 |
| 30 m lower | 0.61 | 0.00 | 0.79 | 1.40 |

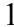

Figure 1. Map of the study area showing Lake Palcacocha and the city of Huaraz in the Quillcay

watershed and the Digital Elevation Model (DEM) of Quillcay watershed. The locations where

hydrographs of the FLO2D simulation results are illustrated are marked as cross-sections.

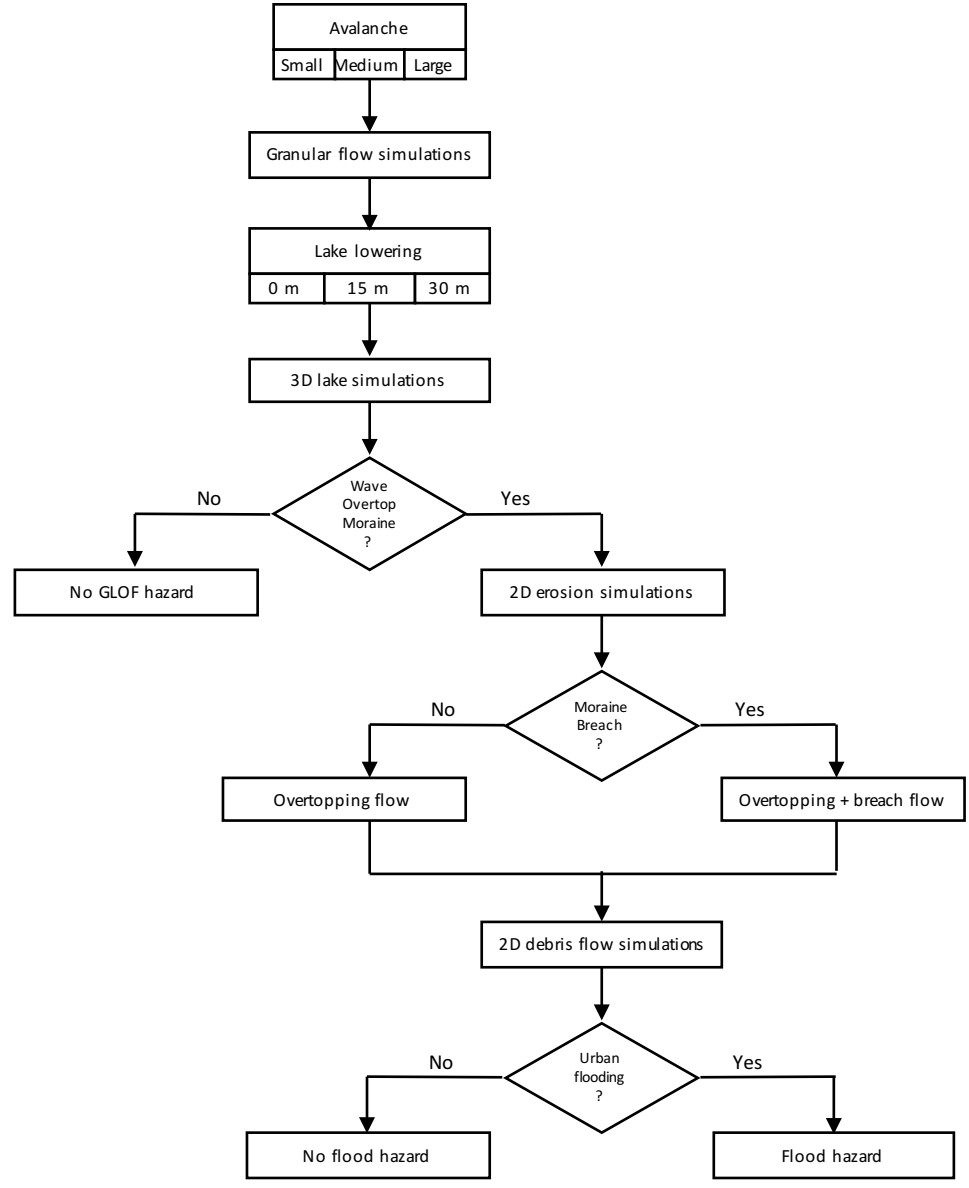

Figure 2. Flowchart of the hazard process chain for an avalanche triggered GLOF from a glacial
lake to assess potential downstream inundation.

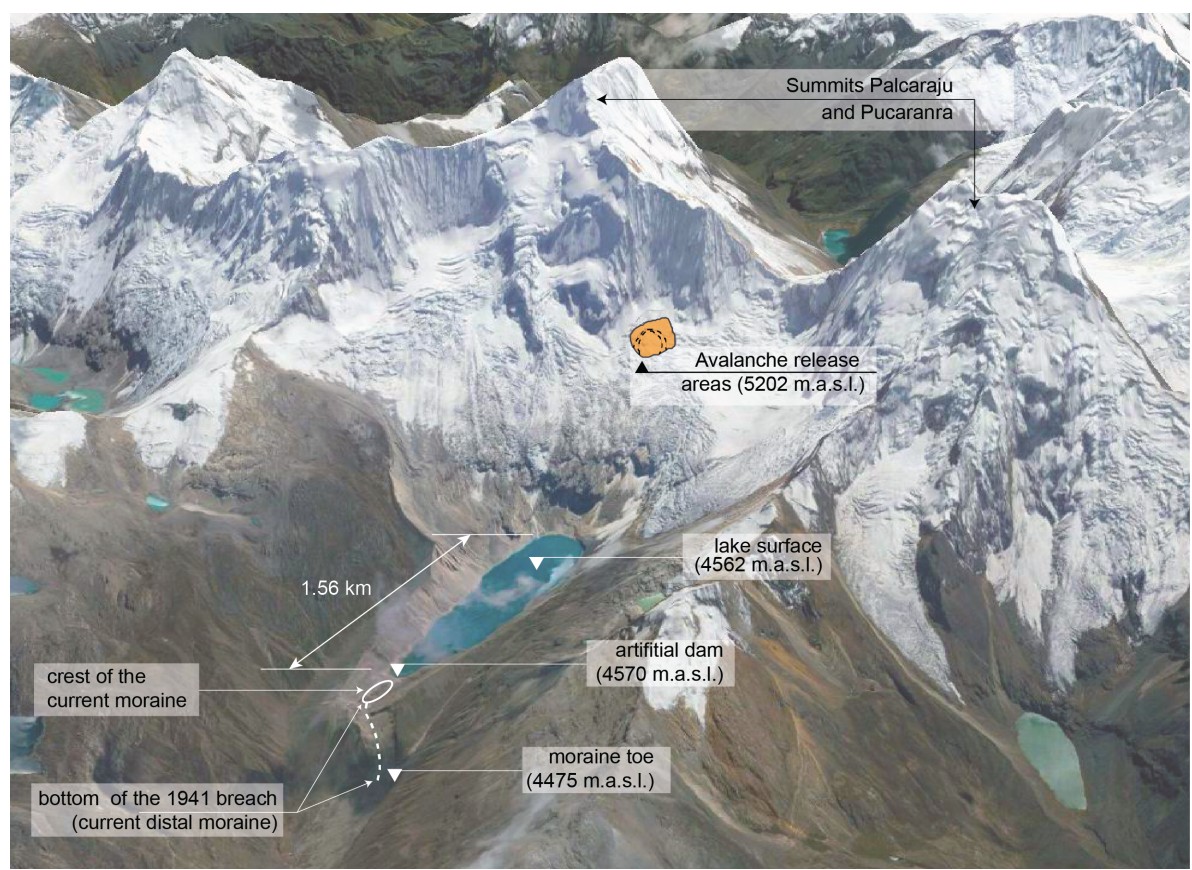

Figure 3. Lake Palcacocha in 2014 with Palcaraju (6,274 m) on the left and Pucaranra (6,156 m) on the right in the background and the 1941 GLOF breach below the lake. Potential avalanche release areas located at an elevation of 5202 m to the north east of Lake Palcacocha following the main axis of the lake. (Google Earth, 2014).

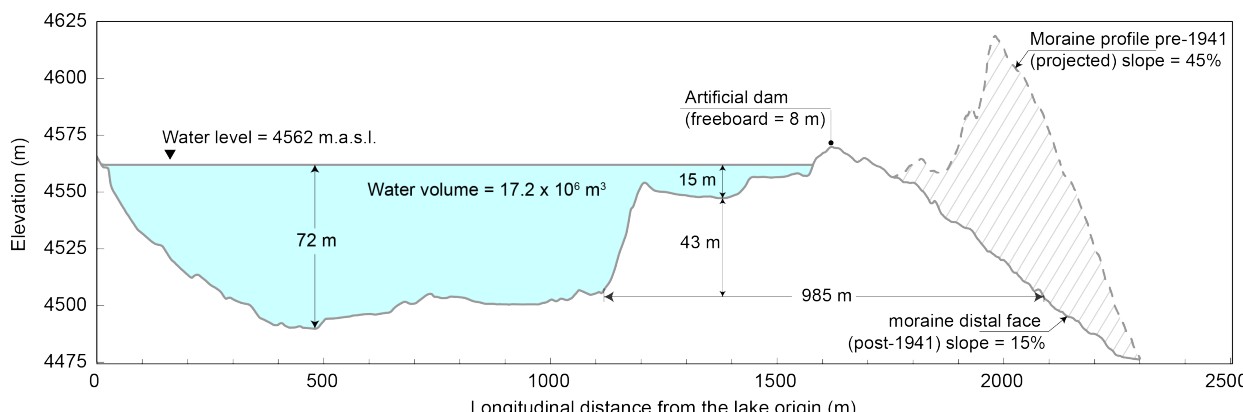

Figure 4. Longitudinal profile of Lake Palcacocha and its terminal moraine (factor of vertical exaggeration of 5). The moraine profile before the 1941 GLOF exhibited width-to-height ratios of 6, while the reshaped moraine after 1941 shows width-to-height ratios of 14 and gentler slopes of 15% (after Rivas et al., 2015).

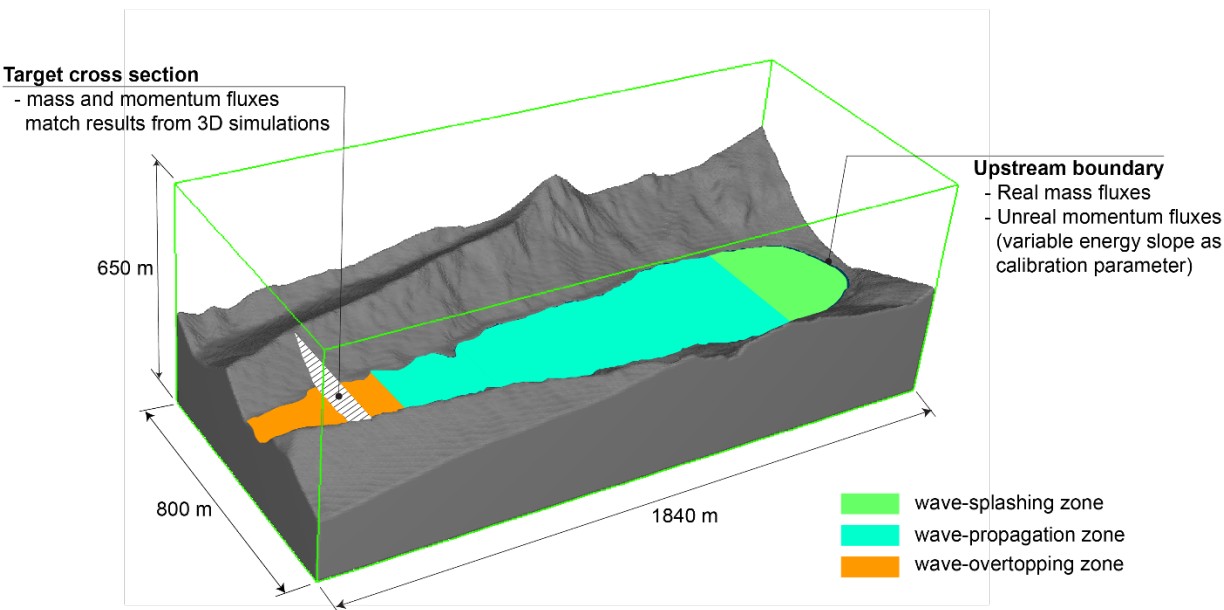

Figure 5. Zones of comparison to validate using BASEMENT for wave-driven breach models. The length of each zone is conceptual and not precise. The locations of the upstream boundary and the target cross section coincide with equivalent flux surfaces in FLOW3D.

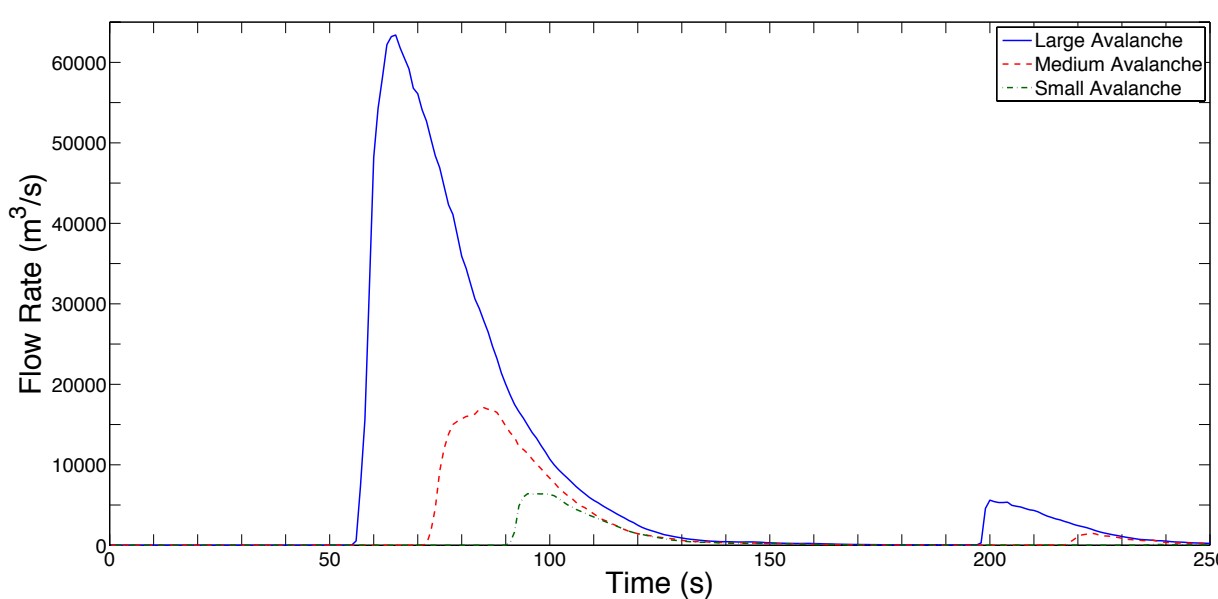

Figure 6. Overtopping wave discharge hydrographs for the three avalanche events with the lake
at its current level.

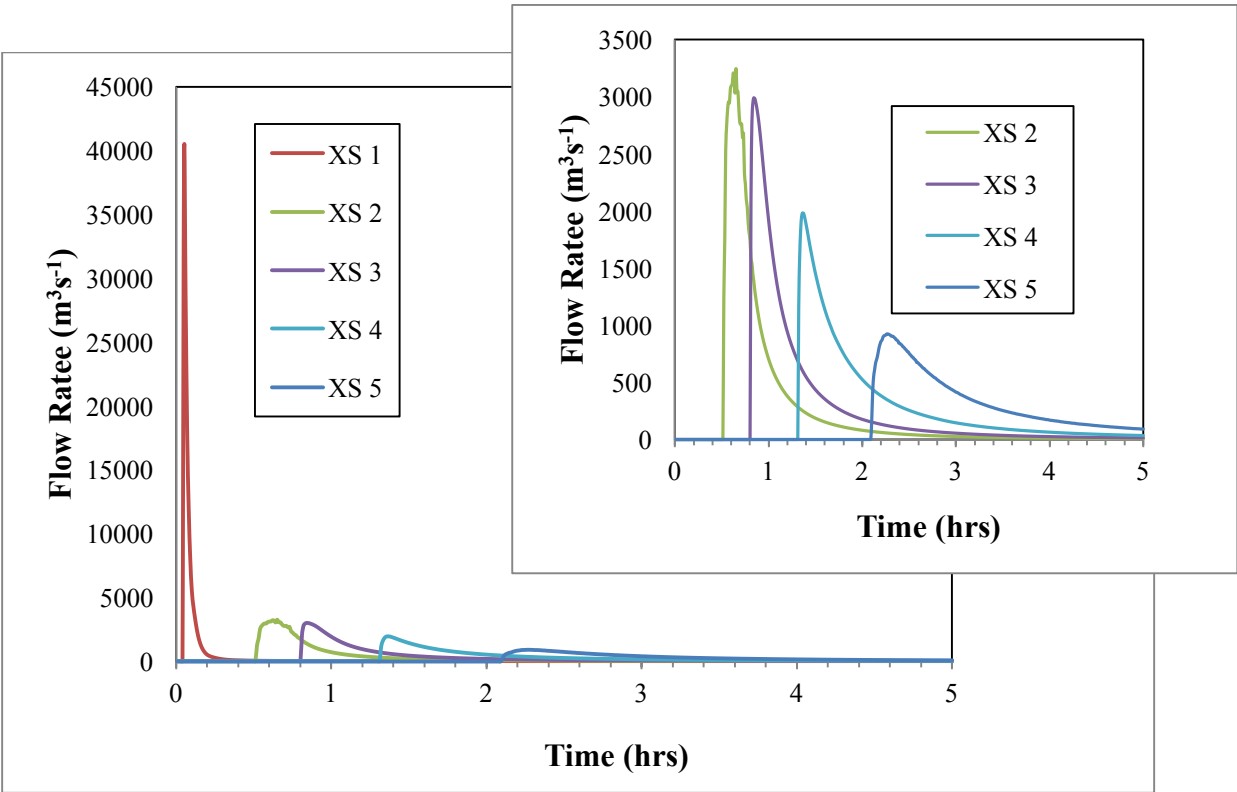

Figure 7. Flood hydrographs at 5 cross-sections downstream of Lake Palcacocha for the large avalanche and current lake level scenario.  Inset shows results on a larger vertical scale for cross-sections 2-5.

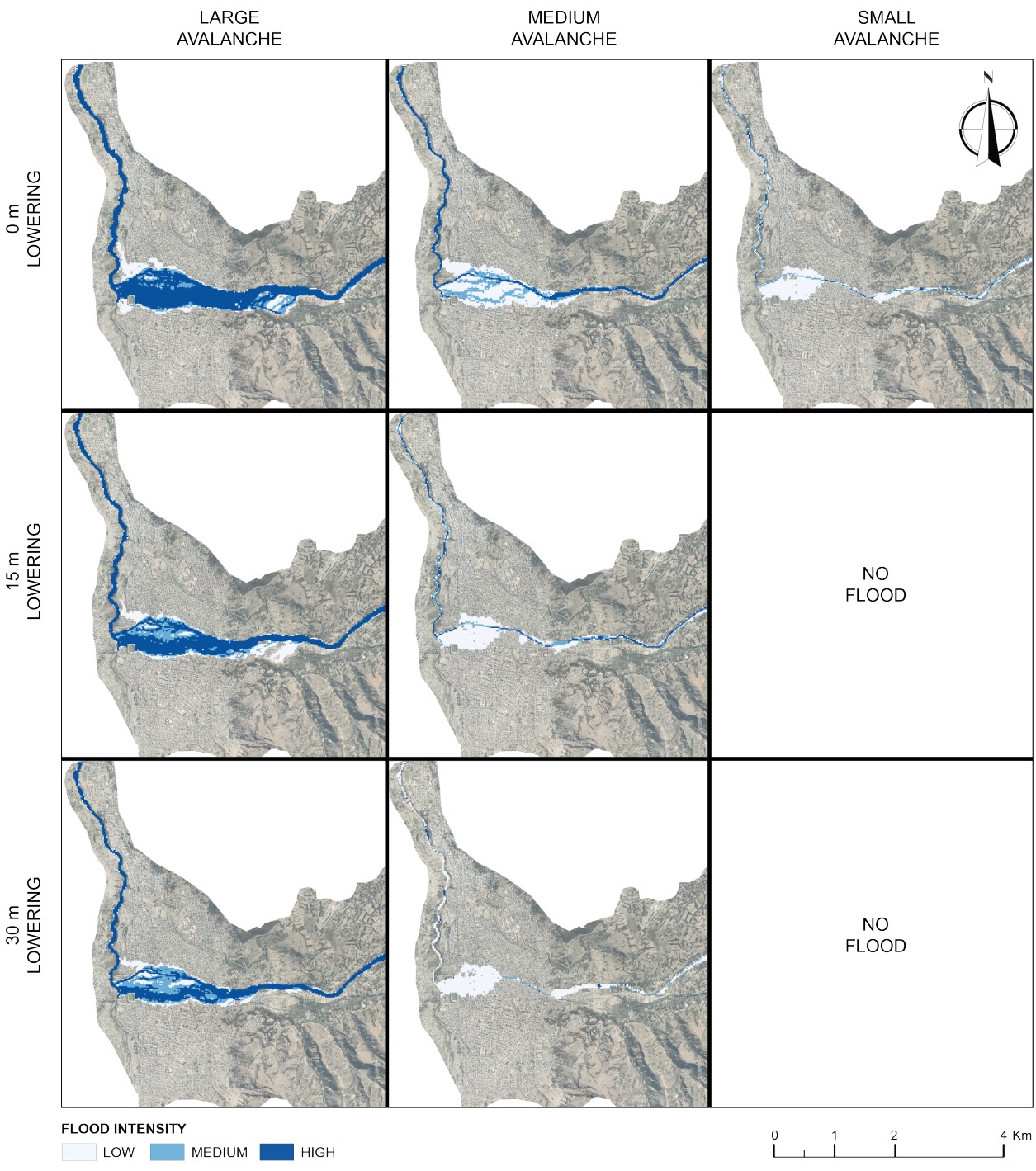

Figure 8. Flood intensity in Huaraz associated with a potential GLOF from Lake Palcacocha for scenarios of 0 m of lake lowering (current condition), 15 m lowering and 30 m lowering conditions for small, medium and large avalanches.

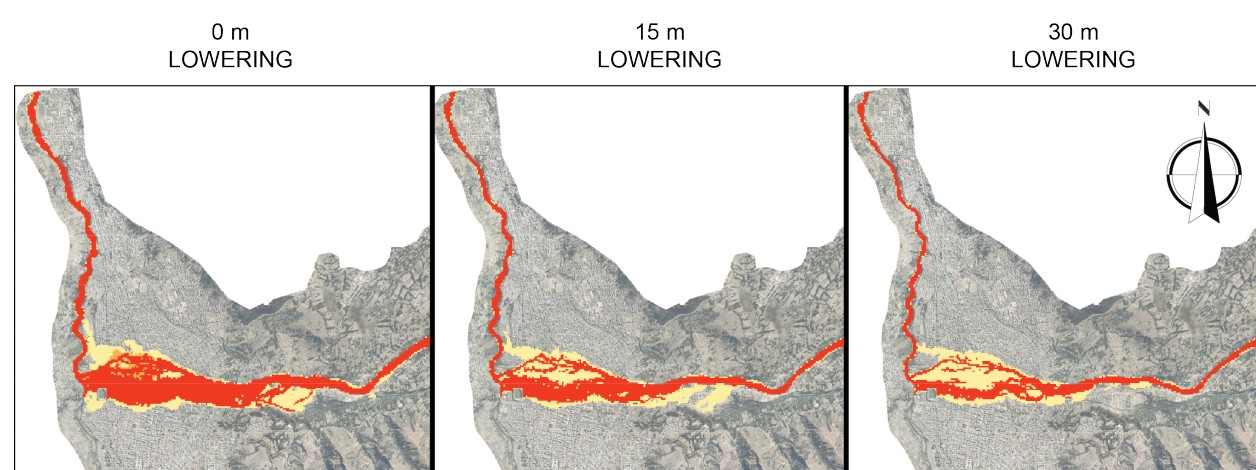

0 m
LOWERING

15 m
LOWERING

30 m
LOWERING

FLOOD HAZARD

LOW  MEDIUM  HIGH

1   2   4 Km
Figure 9. Preliminary hazard map of Huaraz due to a potential GLOF originating from Lake
Palcacocha with the lake at its current level (0 m lowering) and for the two mitigation scenarios
(15 m lowering, and 30 m lowering).

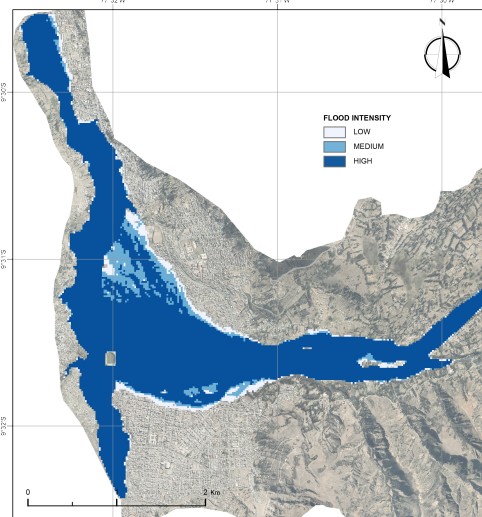

Figure 10. Flood intensity in Huaraz associated with a probable maximum inundation GLOF
from Lake Palcacocha for the scenario of 0 m lake lowering condition and a large avalanche.

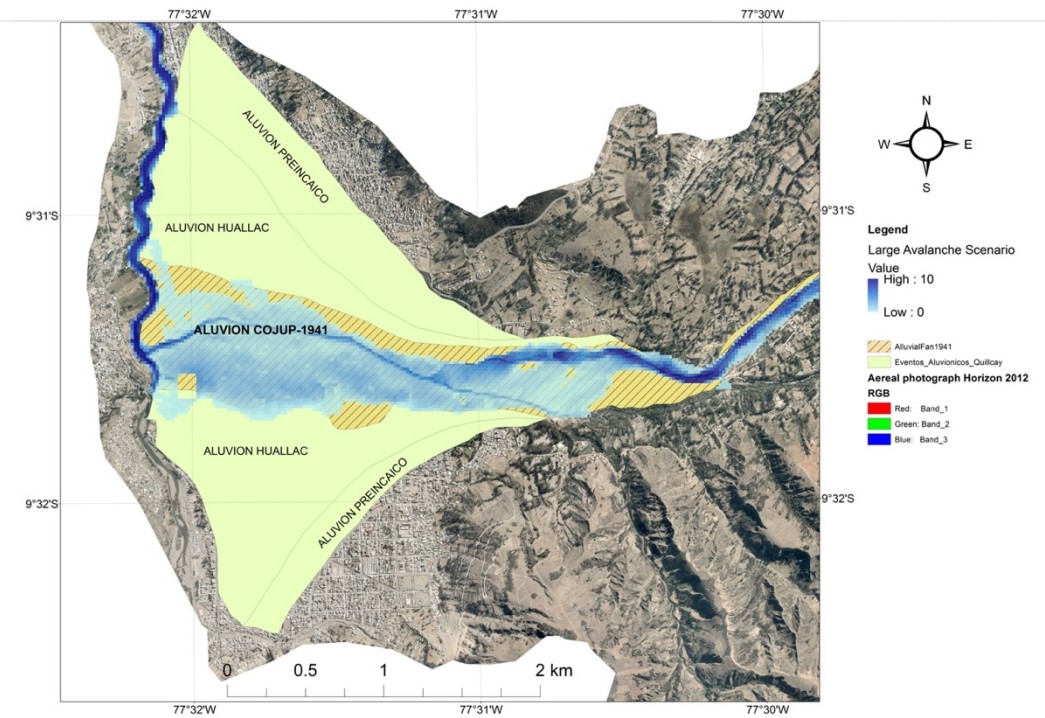

3  Figure 11. Maps published by INDECI (2003) indicating the extension of past mudflow events

4  with the large avalanche scenario superimposed on top of them.

