# Peer review of "Modeling Glacial Lake Outburst Flood Process Chain: The"

_Hydrology and Earth System Sciences, 2015_

## Referee Comment (RC1) · C. Huggel (Referee) · 15 Feb 2016

General comments:

This paper presents a modeling approach for glacier lake outburst floods in Peru, considering the multiple processes from ice avalanche flow, to lake impact, wave generation, lake dam overtopping, erosional process at the moraine dam and flood propagation downstream, including into the Huaraz urban area. There exist currently only very few studies that have developed and applied a similar approach to such multiple cascading mass flow processes. This papers offers important progress on several aspects, is well and comprehensively written and I would like to see this paper published after revisions. In the following I list a few more general points which I think need to be

considered, and then follow on with more specific points. Most of the points are rather minor but there are a few that have more a major character.

The coupling of different, existing (partly commercial) models to mimic the cascading processes is among the most novel and interesting aspects of this paper. The interface of the models, with corresponding output/input, however, is not always crystal clear and I made some specific comments at the respective point in the ms where I think the text has to be more clear.

Use of past events to calibrate the models: there was a GLOF in 1941 from the same lake which unfortunately has never been studied in any reasonable detail so far. The breaching process at the lake was different than what today could happen because the moraine at that time was intact. So not of much use for the lake overtopping / dam breach process but there is probably interesting information out there in terms of flood propagation and flood levels and flow characteristics down to Huaraz. I'm definitely not asking for a detailed study and comparison of this historical case, this would be way beyond the scope of this paper (also considering that the paper already is quite packed). However, I think the authors should make reference to this event and the potential to compare or calibrate the flood models applied. I recently visited the area and there are some flood deposits visible in the flow channel (in the Cojup valley) which may be used to calculate discharge per cross-section. Or they may want to use the historical photographs available to compare their inundation areas with the historical case (at least in a qualitative sense).

I note that the authors do not consider the scenario of a moraine collapse as occurred in 2003, producing a (relatively small) overtopping and downstream flood. This should be discussed.

The effect of flow transformation downstream of the lake with different flow rheologies (e.g. from debris flow to hyperconcentrated flow and back to debris flow) has not been considered but is likely to occur and probably important, e.g. for travel times. The

authors should at least discuss possible effects and limitations in their model setting with respect to this process.

The uncertainties of the models and their propagation through the models is not well assessed or discussed. This should be included. The authors may want to consider a new publication on this, on the example of the Lake 513 nearby: Schaub et al., Landslides, 2016. I think the authors should make a statement concerning the robustness of their model results (especially in terms of the final inundation maps).

An important point concerning the hazard map: a hazard map should never be a direct result of a model output. Field evaluation and validation is an essential part of a hazard map. I suggest to call the map a 'preliminary hazard map', making reference to the importance of field evaluation (which could not be done for this paper).

Overall, the paper is well written and I really appreciate the comprehensive literature review and the methodological details which help any reader to better follow and understand. However, I think there is a bit of redundancy here and there.

Specific comments

Page 5, lines 3-8 is redundant

p. 7, l. 23: do you have evidence of increased frequency of extreme precipitation? I did not see any study on this so far.

p. 9, l. 5: I think this should be hazard rather than risk assessment

p. 10, l. 8: I suggest to explicitly state the type of avalanche p. 10, l. 28: slab failures can also be produced at larger glaciers

p. 11, l. 7ff: there is an important mis-understanding here that needs to be corrected. The formula of Huggel et al 2004 relates avalanche volume to average slope of the runout (i.e. from the point of failure to the furthest point of runout), and NOT to the slope of the failure surface/glacier!

[Figure]

p. 13: the interface of RAMMS avalanche model, and FLOW-3D could be described somewhat more explicitly.

p. 14, l. 2: I suggest to change conservative risk perspective into worst-case approach

p. 16: the interface of FLO-3D and BASEMENT should be described more clearly and in terms of the calibrated parameters. Also, where exactly is BASEMENT started? I think an additional table with the parameters could help.

p. 18, 18: Actually, not many models are currently capable of simulating entrainment processes, most examples mentioned are not.

p. 20, l. 7: the area reduction factor could probably also be higher than 20%, considering the building density in Huaraz.

p. 20, l. 23ff: according to Table 2, the intensity matrix for floods and not for debris flows (of the Swiss system) is applied. The model simulates debris flow, so the debris flow intensity levels may be more appropriate.

p. 21, l. 17ff: I see a need to extend how hazard zones were mapped. As mentioned above, a direct conversion of model output to a hazard map is not appropriate (preliminary hazard map may be more appropriate here).

p.22/23: I suggest to include the results of the comparison with the Heller and Hager model in Table 4. This is of interest.

p. 24: I found the evaluation of different lake lowering scenarios particularly useful from an engineering point of view and represents a work that is hardly done.

p. 25, l. 20ff: I'm not whether failure is the best term here because it may be ambiguous in a case where a breached moraine already exists. I'd rather use full breach development, implying that the lake drains completely. Please clarify this.

p. 26, l. 27/28: Almost one hour to cross the urban area seems high to me for a GLOF. Please check whether you may need to adjust the FLO-2D model parameters for the

urban areas.

p. 27, l. 1-6: I would be good to also show the arrival times for the small/medium scenarios (cf also Fig. 7).

p. 27 (4.6): The decision which scenario to eventually include in a hazard map is also a political and just a scientific question. I would explicitly mention this. To me, the approach taken seems reasonable. We have discussed this issue in a workshop in Huaraz (with participation of Rachel Chisolm, a co-author of this paper). There was not a clear opinion or statement on this. I think assessing the worst-case is something science should do, and its inclusion in terms of a residual hazard zone seems reasonable to me (considering that all hazard zones presented here should be labeled preliminary).

p. 28, l. 21ff: data on past events is available (ie the 1941 GLOF), at least for the downstream mass flow, and this should be discussed, as previously mentioned.

p. 29, l. 9-11: I agree that the use of a 3D model is adding value to the assessment of lake displacement waves and is likely to capture the complexity better than simpler models. However, I don't quite agree with this statement which seems to me to be overly confident with this model. Overall, there is only limited experience with this kind of model for such environments and there is substantial number of model parameters to be calibrated. I suggest to discuss the uncertainties that are related to this model.

p. 30, l. 6: I suggest to use worst-case instead of conservative approach.

p. 32, l. 10: I guess you are talking about hazards since the paper does not contain any material on risk.

Figures are of good quality and I particularly like Fig. 7. Table 8 can probably be avoided.

Christian Huggel, University of Zurich.

---

## Referee Comment (RC2) · Anonymous Referee #2 · 24 Mar 2016

Modeling glacial lake outburst flood process chain: the case of Lake Palcacocha and Huaraz, Peru. Referee comments The paper describes multi-scenario modeling of glacial Lake Palcacocha outburst and inundation of Huaraz city. The parameters of the possible GLOF are evaluated on the base of the chain of several models. Such approach gave possibility to authors to carry out potential hazard mapping of Huaraz city, which due to its location is very vulnerable to mountain disasters. The paper is good structured; each part of modeling is described indetail and enough illustrated. This research is very important for the developing of mitigation measures according Lake Palcacocha possible GLOF.

Some small improvements could be done before publication: Abstract, line 8

and Study area,p.7 line 25. There are different estimations of number of victims during catastrophic GLOF in 1941 exist. For example, Mark Carey (Mark Carey, In the Shadow of Melting Glaciers: Climate Change and Andean Society, 2010, DOI:10.1093/acprof:oso/9780195396065.003.0002) wrote, that "glacial lake outburst flood in 1941 killed 5,000 people and destroyed one-third of the Ancash capital city of Huaraz". So, it is may be better to give several references in one place (for example, in the Study area description). P.8 Study area. As shown on the fig.1, there are several other river branches with lakes in the area above Huaraz city. Does any possibility of their outburst exist? Or Lake Palcacocha is only one potentially dangerous lake in the basin? It could be interesting to the reader. P. 14 Moraine erosion simulation. It is not rare case in the glaciated areas, when moraine dam contains ice or frozen patterns. In such case dam erosion process during outburst flood has other mechanism and erosion can be larger. Whether the damming moraine of Lake Palcacocha may contain ice? This point should be mentioned and discussed. P.18. Inundation simulation. FLO-2D model chosen for inundation simulation, doesn't take into account additional erosion and subsequent accumulation of debris during flood wave moving. However, there are several zones of erosion and accumulation of debris from 1941 GLOF along the Paria River, and the same additional erosion and accumulation could be expected for the next GLOF event. Such redeposition is very difficult take into account during modelling, but this model limitation should be mentioned. P.21 3.6 Hazard identification. To my opinion, it is better to use term "potential hazard" instead "hazard" for described hazard zonation. P.4 line 10 Fischer et al., 2012) –left parenthesis is missed

Please also note the supplement to this comment:
http://www.hydrol-earth-syst-sci-discuss.net/hess-2015-512/hess-2015-512-RC2-supplement.pdf

---

## Author Response (AR1)

**HESS-2015-512-Discussions**

Modeling glacial lake outburst flood process chain: the case of Lake Palcacocha and Huaraz, Peru

M. A. Somos-Valenzuela, R. E. Chisolm, D. S. Rivas, C. Portocarrero, and D. C. McKinney

**RESPONSE TO THE COMMENTS OF REVIEWER 1**

The authors greatly appreciate the insightful and constructive comments of Dr. Christian Huggel that helped us to improve the paper.

**GENERAL COMMENTS**

**General Comment 1:**

Use of past events to calibrate the models: there was a GLOF in 1941 from the same lake which unfortunately has never been studied in any reasonable detail so far. The breaching process at the lake was different than what today could happen because the moraine at that time was intact. So not of much use for the lake overtopping/dam breach process but there is probably interesting information out there regarding flood propagation and flood levels and flow characteristics down to Huaraz. I'm definitely not asking for a detailed study and comparison of this historical case, this would be way beyond the scope of this paper (also considering that the paper already is quite packed). However, I think the authors should make reference to this event and the potential to compare or calibrate the flood models applied. I recently visited the area and there are some flood deposits visible in the flow channel (in the Cojup valley) which may be used to calculate discharge per cross-section. Or they may want to use the historical photographs available to compare their inundation areas with the historical case (at least in a qualitative sense).

- **Response to General Comment 1:**

We agree with the reviewer that a study of the GLOF event from 1941 would add a lot of information for this work and similar works elsewhere. Although the reviewer pointed out correctly that such effort is out of the scope of this particular manuscript, it is important to mention the authors attempt in 2012 to carry out such studies in the area with the help of the Mountain Institute in Huaraz, the Ministry of Environment of Peru and the support of the Interamerican Development Bank. Our goal then was to generate a high resolution DEM, study the GLOF from 1941, the stability of the moraine and the debris that a potential GLOF could pick up on its way to Huaraz. Unfortunately all of this could not be completed and we were only able to finance the generation of the DEM which is used in this study. Additionally, the 1941 event changed the topography, so it is not completely analogous to the potential event we are modeling (Rivas et al., 2015). The qualitative comparison described in the next paragraphs has been added to the end of the Discussion section of the revised paper.

New text: "There are still many unknowns about the 1941 event, including the precise lake volume at that time, underlying bathymetry and pre-GLOF moraine morphology, flood volume and discharge hydrograph; aerial images and derived historical maps represent the only sources of information, known to the authors, about the pysical characteristics of the 1941 GLOF, providing at least a rough visual estimation of the flood area. In a qualitative comparison with the GLOF from 1941, we used a map published by the Instituto Nacional de Defensa Civil (INDECI, 2003) where three mudflow event extensions are delineated: Aluvion Preincaico, Aluvion Huallac and Aluvion Cojup 1941. In Figure 1 (Figure 11 in revised paper) we plot the inundation extension reported in this paper on the map of the 1941 event delineated by INDECI (2003) and confirm that the inundation modeled has reasonable dimensions in comparison with this historical information. The volume at the time was estimated to be on the order of 14 million m$^3$ (Vilímek et al., 2005), which is more than 7 times the volume that we have calculated for the large avalanche (1.8 million m$^3$). This may explain the fact that in our results the inundation does not pass out of the bank from the Cojup River to the Quilcaihuanca River in the area where the rivers are very close together near the entrance to the eastern border of the city. However, these results demand caution; a qualitative comparison only describes potential differences between simulated and observed flood areas. Because the moraine failure in 1941 changed the upstream conditions at Lake Palcacocha, historical aerial images of flooded areas constitute no source of information for precise calibration for our model."

[Figure]

Figure 1 (Figure 11 in revised paper). Maps published by INDECI (2003) indicating the extension of past mudflow events with the large avalanche scenario superimposed on top of them.

INDECI – Instituto Nacional de Defensa Civil, Plan de Prevención ante Desastres: Usos del Suelo y Medidas de Mitigacion Ciudad de Huaraz. Plate 33, Proyecto INDECI – PNUD PER/02/051 Ciudades Sostenibles, Lima, 2003.

http://bvpad.indeci.gob.pe/doc/estudios_CS/Region_Ancash/ancash/huaraz.pdf (Accessed April 15, 2016)

Vilímek, V., M.L. Zapata, J. Klimeš, Z. Patzelt, and N. Santillán, 2005. Influence of Glacial Retreat on Natural Hazards of the Palcacocha Lake Area, Peru. Landslides 2:107–115.

**General comment 2:**

I note that the authors do not consider the scenario of a moraine collapse as occurred in 2003, producing a (relatively small) overtopping and downstream flood. This should be discussed.

- **Response to General Comment 2:**

We agree that this should be discussed and the following text has been added to the Discussions section of the revised paper:

New Text: "According to Vilímek *et al.* (2005), the lateral moraine collapse that occurred in 2003 at Lake Palcacocha was due to a wave produced by a landslide on the internal face of the left lateral moraine that was triggered by extensive rainfall precipitation which over-saturated the moraine material. The terminal moraine was eroded but it did not breach. A downstream flood was produced by the water that overtopped the moraine. While this type of landslide from the lateral moraine is likely to occur again in the future, the work reported here focuses on the potential effects of an avalanche-generated wave because the magnitude of landslides likely to enter the lake are less than the avalanche volumes we have considered, and the effect of a landslide-generated wave may be somewhat mitigated as it propagates diagonally across the lake, whereas an avalanche-generated wave would enter along the longitudinal axis of the lake and is unlikely to be attenuated by reflections off of the lateral moraines.

Even though a prescribed terminal moraine collapse scenario was simulated, it was not included in the preliminary hazard map for two reasons. First, the complete collapse scenario is based on the premise that we should consider a worst case scenario, but we could not initiate the moraine collapse using our numerical approach; even when a large overtopping wave and highly erosive materials were assumed, the width of the moraine is simply too great, and the erosion does not extend from the distal face of the moraine back to the lake. Therefore, we artificially prescribed and simulated the moraine collapse. Using empirical equations we determined the time that the collapse will take and the hydrograph was calculated following hydrodynamic constraints as indicated in Rivas *et al.* (2015). Based on these modeling results it is extremely unlikely that the collapse will occur, but it cannot be completely disregarded. Secondly, given the magnitude of the extremely unlikely breach scenario results, it is important to avoid creating confusion as a result of misinterpretation of the results. People in Huaraz should decide if they want to consider the worst case scenario in their planning, and this work is limited to informing that decision making process."

Rivas, D. S., Somos-Valenzuela, M. A., Hodges, B. R., and McKinney, D. C.: Predicting outflow induced by moraine failure in glacial lakes: The Lake Palcacocha case from an uncertainty perspective, Nat. Hazards Earth Syst. Sci., 15, 1163-1179, 2015.

Vilímek, V., M.L. Zapata, J. Klimeš, Z. Patzelt, and N. Santillán, 2005. Influence of Glacial Retreat on Natural Hazards of the Palcacocha Lake Area, Peru. Landslides 2:107–115.

**General comment 3:**

The effect of flow transformation downstream of the lake with different flow rheologies (e.g. from debris flow to hyper concentrated flow and back to debris flow) has not been considered but is likely to occur and probably important, e.g. for travel times. The authors should at least discuss possible effects and limitations in their model setting with respect to this process.

- **Response to General Comment 3:**

We agree with this comment, especially given the importance of the problem analyzed. With regard to the possible effects and limitations in the model setting with respect to different flow rheologies, we identified two major sources of uncertainty: (1) the physical characteristics of the mixture and (2) the volume of material that will be eroded, transported and deposited again, a process that may happen many times during the trajectory of the flood. FLO2D can simulate the behavior of the mixture assuming that it won't change throughout the simulation. Consequently, it is not able to consider transformations of the flow rheology except for changes in concentration by volume that can change the dynamic viscosity ($\eta$) and yield stress ($\tau_y$), where

$$\eta = \alpha_1 e^{\beta_1 C_v} \tag{1}$$

$$\tau_y = \alpha_2 e^{\beta_2 C_v} \tag{2}$$

where $\alpha_i$ and $\beta_i$ are empirical coefficients defined by laboratory experiment and $C_v$ is the sediment concentration by volume (O'Brien and Julien, 1988). Additionally, scouring is not simulated in the FLO2D mudflow module, so we prescribe the concentration by volume to be 50% based on the literature recommendations.

The quadratic rheological model used within FLO2D combines four stress components of hyper-concentrated sediment mixtures: (1) cohesion between particles; (2) internal friction between fluid and sediment particles; (3) turbulence; and (4) inertial impact between particles, where the cohesion between particles is the only parameter that is independent of the mixture concentration or hydraulic characteristics (Julien, 2010:243; O'Brien and Julien, 1988). According to the few studies of the composition of the Lake Palcacocha moraine (Novotný and Klimeš, 2014, section 3.3), the cohesion can be considered nearly equal to zero, which implies that the resulting mixture would have low yield stress and dynamic viscosity. Consequently, from the list of 10 soils presented in the FLO2D manual (FLO2D, 2012: Table 8, p. 57), we selected parameters that give a low yield stress and dynamic viscosity (Glenwood 2 from Table 1 below). In addition, a sensitivity analysis was performed using the parameters for the other soils listed in Table 1 (Aspen Pit 2, Glenwood 1, and Glenwood 3 with higher dynamic viscosities and yield stresses, and Glenwood 4 with much higher values). The results of the sensitivity analysis (FLO2D simulations) show that the flood arrival time at cross section 4 (see Figure 1 in the original paper) varies from 1.05 to 1.32 hours (compared to 1.32 hours with Glenwood 2 parameters, see Table 6 in original paper). The peak flow varies from 1954 to 3762 $m^2s^{-1}$ (compared to 1,980 $m^3s^{-1}$ using Glenwood 2). The Glenwood 4 parameters result in the shorter arrival time and higher peak value. Therefore, the rheology, which is a function of the concentration of the mixture and the soil characteristics, does affect the travel time and the peak flows. The results are not expected to be highly sensitive if the dynamic viscosity were to be lower than what was assumed (Glenwood 2), which is expected from the few soil studies in the area.

Table 1: Yield Stress ($\tau_y$) and Dynamic Viscosity ($\eta$) as a Function of Sediment Concentration (adapted from FLO2D, 2012)

| Source | Yield Stress ($\tau_y$ ) | | | Dynamic Viscosity ($\eta$) | | |
|---|---|---|---|---|---|---|
| | $\alpha_2$ | $\beta_2$ | $\tau_y$ (dynes $cm^{-2}$) | $\alpha_1$ | $\beta_1$ | $\eta$ (poises) |
| Aspen Pit 2 | 2.72 | 10.4 | 493 | 0.0538 | 14.5 | 76 |
| Glenwood 1 | 0.0345 | 20.1 | 799 | 0.00283 | 23 | 279 |
| Glenwood 2 | 0.0765 | 16.9 | 358 | 0.0648 | 6.2 | 1 |
| Glenwood 3 | 0.000707 | 29.8 | 2091 | 0.00632 | 19.9 | 132 |
| Glenwood 4 | 0.00172 | 29.5 | 4379 | 0.000602 | 33.1 | 9272 |

New Text: "With regard to the possible effects and limitations in the model setting with respect to different flow rheologies, we identified two major sources of uncertainty: (1) the physical characteristics of the mixture and (2) the volume of material that will be eroded, transported and deposited again, a process that may happen many times during the trajectory of the flood. FLO2D can simulate the behavior of the mixture assuming that it won't change throughout the simulation. Consequently, it is not able to consider transformations of the flow rheology except for changes in concentration by volume that can change the dynamic viscosity ($\eta$) and yield stress ($\tau_y$) (O'Brien and Julien, 1988). Additionally, scouring is not simulated in the FLO2D mudflow module, so we prescribe the concentration by volume to be 50% based on the literature recommendations.

The quadratic rheological model used within FLO2D combines four stress components of hyper-concentrated sediment mixtures: (1) cohesion between particles; (2) internal friction between fluid and sediment particles; (3) turbulence; and (4) inertial impact between particles, where the cohesion between particles is the only parameter that is independent of the mixture concentration or hydraulic characteristics (Julien, 2010:243; O'Brien and Julien, 1988). According to the few studies of the composition of the Lake Palcacocha moraine (Novotný and Klimeš, 2014), the cohesion can be considered nearly equal to zero, which implies that the resulting mixture would have low yield stress and dynamic viscosity. Consequently, from the list of 10 soils presented in the FLO2D manual (FLO2D, 2012: Table 8, p. 57), we selected parameters that give a low yield stress and dynamic viscosity (Glenwood 2). In addition, a sensitivity analysis was performed using the parameters for the other soils listed in Table 1 (Aspen Pit 2, Glenwood 1, and Glenwood 3 with higher dynamic viscosities and yield stresses, and Glenwood 4 with much higher values). The results of the sensitivity analysis (FLO2D simulations) show that the flood arrival time at cross section 4 (see Figure 1 in the original paper) varies from 1.05 to 1.32 hours (compared to 1.32 hours with Glenwood 2 parameters, see Table 6 in original paper). The peak flow varies from 1954 to 3762 $m^2s^{-1}$ (compared to 1,980 $m^3s^{-1}$ using Glenwood 2). The Glenwood 4 parameters result in the shorter arrival time and higher peak value. Therefore, the rheology, which is a function of the concentration of the mixture and the soil characteristics, does affect the travel time and the peak flows. The results are not expected to be highly sensitive if the dynamic viscosity were to be lower than what was assumed (Glenwood 2), which is expected from the few soil studies in the area."

FLO2D: FLO2D PRO Reference Manual, FLO2D Software, Inc., Nutrioso, AZ, 2012.

Julien, P. Y.: Erosion and Sedimentation, second edition, Cambridge, UK: Cambridge University Press, 371 pp., 2010.

Novotny, J. and Klimes, J.: Grain size distribution of soils within the Cordillera Blanca, Peru: an indicator of basic mechanical properties for slope stability evaluation, J. Mount. Sci., 11, 563–577, 2014.

O'Brien, J.S. and P.Y. Julien, 1988. Laboratory Analysis of Mudflow Properties. Journal of Hydraulic Engineering 114:877–887.

**General comment 4:**

The uncertainties of the models and their propagation through the models is not well assessed or discussed. This should be included. The authors may want to consider a new publication on this, on the example of the Lake 513 nearby: Schaub et al., Landslides, 2016. I think the authors should make a statement concerning the robustness of their model results (especially in terms of the final inundation maps).

- **Response to General Comment 4:**

A complete uncertainty analysis of the hazard process chain modeled here is beyond the scope of this paper, but it would make an interesting new paper building on this work. We agree that a qualitative discussion of the uncertainties in each modeled process would improve this paper, and we have incorporated this into the discussion section of the paper.

New text: "For the sensitivity analysis of the inundation, we focused our effort on three components: (1) sediment concentration by volume, (2) roughness, and (3) rheology of the flow. The concentration by volume and roughness were analyzed in the dissertation of Somos-Valenzuela (2014) who concluded that concentration is not a main factor affecting travel time, but it does affect the downstream inundated area in Huaraz, since the volume of the flow increases as concentration increases. However, travel time is sensitive to roughness, increasing to 1.5 hours at cross section 4 (compared to 1.32 hours for the baseline simulation) as the roughness coefficient is increased from 0.1 to 0.4. Also, the peak flow is inversely proportional to the roughness, so lower roughness results in a slightly higher peak (less than 10%) (Somos-Valenzuela, 2014). For the rheology parameters, as noted above, the FLO2D model was run using a different set of parameters and it was found that travel time increased as the dynamic viscosity was increased, and the same is true for the peak flow.

Considering the robustness of the inundation map, of all the parameters analyzed, we found that the parameter that most influences the inundated area is the volume of the inundation, which is a combination of the flow released from the lake and the material picked up along the way to Huaraz. Coincidentally, the size of the flood is one the most difficult parameters to calculate, which is a consequence of the difficulties in estimating the size of an avalanche that might hit the lake.

The lake model has a considerable amount of uncertainty. The greatest sources of uncertainty are the avalanche characteristics (inputs to the lake model) and the wave generation. The processes associated with wave generation from avalanche impact are poorly understood, and current model limitations do not allow for an avalanche to be simulated with its actual flow characteristics (rheology, density, etc.) in the same environment as the lake dynamics. Therefore, it is difficult to represent wave generation in a fully physical manner. Sensitivity analysis shows that the avalanche characteristics (depth and velocity) have a significant impact on the wave characteristics and moraine overtopping hydrograph. Additionally, the method of representing the avalanche impact boundary condition may overestimate the momentum of the inflow; the result of this may be somewhat larger wave height, but the greatest impact is in the peak flow and total volume of the overtopping wave. The highest estimates of the overtopping wave characteristics are presented in the paper to illustrate a worst-case scenario, but it is likely that the actual magnitude of an avalanche generated wave may be less than what is reported here."

**General comment 5:**

An important point concerning the hazard map: a hazard map should never be a direct result of a model output. Field evaluation and validation is an essential part of a hazard map. I suggest to call the map a 'preliminary hazard map', making reference to the importance of field evaluation (which could not be done for this paper).

**Response to General Comment 5:**

We agree with this suggestion and we have implemented it in the paper.

**General comment 6:**

Overall, the paper is well written and I really appreciate the comprehensive literature review and the methodological details which help any reader to better follow and understand. However, I think there is a bit of redundancy here and there.

**Response to General Comment 6:**

We agree with this suggestion and we have implemented it in the paper.

**SPECIFIC COMMENTS**

**Specific Comment 1:**

Page 5, lines 3-8 is redundant

- **Response to Specific Comment 1:**

We agree and have modified the text by deleting this sentence and moving the reference to p.4 – L5-15.

[revised manuscript text omitted]

**Specific Comment 2:**

p. 7, l. 23: do you have evidence of increased frequency of extreme precipitation? I did not see any study on this so far.

- **Response to Specific Comment 2:**

We agree with this and have deleted the reference to climate change impacts since this is not the focus of this paper.

Old text: "Climate related impacts on the Quillcay basin include rapid recession of glaciers, resulting in increasing scarcity and worsening quality of water, shifting precipitation patterns and increased frequency of extreme precipitation events; however, the danger of a GLOF from Lake Palcacocha is paramount (HiMAP, 2014). A GLOF originating from the lake occurred in 1941, flooding the downstream city of Huaraz, killing 1800 people (according to best estimates) (Wegner, 2014) and destroying infrastructure and agricultural land all the way to the coast (Carey, 2010; Evans et al., 2009)."

New text "The danger of a GLOF from Lake Palcacocha is paramount (HiMAP, 2014). A GLOF originating from the lake occurred in 1941, flooding the downstream city of Huaraz, killing about 1800 people (according to best estimates) (Wegner, 2014) and destroying infrastructure and agricultural land all the way to the coast (Carey, 2010; Evans et al., 2009)."

**Specific Comment 3:**

P. 9, l. 5: I think this should be hazard rather than risk assessment

- **Response to Specific Comment 3:**

We agree and the word "risk" has been changed to "hazard" the relevant locations in the paper.

**Specific Comment 4:**

p. 10, l. 8: I suggest to explicitly state the type of avalanche

- **Response to Specific Comment 4:**

We include the words ice-rocks after the coma.

Old text: "In non-forested areas, avalanches can be generated on slopes of 30–50°, and in tropical areas the critical slope can be even less (Christen et al., 2005; Haeberli, 2013)."

New text: "In non-forested areas, ice-rock avalanches can be generated on slopes of 30-50°, …"

**Specific Comment 5:**

p. 10, l. 28: slab failures can also be produced at larger glaciers

- **Response to Specific Comment 5:**

We agree with this comment and after reviewing the literature cited, the paragraph has been changed.

Old Text: "Huggel et al. (2004) suggest that ice avalanches in slab failures are produced in small and steep glaciers with thicknesses between 30 to 60 m."

New text "Huggel et al. (2004), after Alean (1985), suggest that ice avalanches in slab failures are mainly produced in small and steep glaciers with thicknesses between 30 to 60 m, where they are less frequent in large valley-type glaciers."

**Specific Comment 6:**

p. 11, l. 7ff: there is an important mis-understanding here that needs to be corrected. The formula of Huggel et al 2004 relates avalanche volume to average slope of the runout (i.e. from the point of failure to the furthest point of runout), and NOT to the slope of the failure surface/glacier!

- **Response to Specific Comment 6:**

We agree with the reviewer and the information that we reported was not clear and we misunderstood equation 5 from Huggel *et al.*, (2004). We really appreciate that the reviewer checked this and pointed it out. Therefore the paragraph on page 11 from line 7 to 17 has been changed:

Old text: "Huggel et al. (2004, Eq. 5) derived a regression equation between average glacier slope (tan α) and avalanche volume from observations of large ice avalanches worldwide. The terrain in the avalanche source areas above Lake Palcacocha has slopes between 20–35° at elevations of 5000–5300m. The regression equation leads to a volume of almost 3x $10^6$ m$^3$ when evaluated for a slope of 20° and 0.5 x $10^6$ m$^3$ for 25°. The slopes above 5300m are greater than 35°, so avalanches originating from higher elevations are expected to be smaller. Three avalanche volumes are considered in this work, 0.5 x $10^6$ m$^3$ (small), 1.0 x $10^6$ m$^3$ (medium) and 3.0 x $10^6$ m$^3$ (large). These potential avalanche volumes are consistent with the elevations and slopes of the source area. The release area (shown in Fig. 2) was located at an elevation of 5200m to the north east of the lake following the main axis of the lake."

New text: **"**Three avalanche volumes are considered in this work, similar to the avalanche scenarios in Schneider et al. (2014): 0.5x$10^6$ m$^3$ (small), 1x$10^6$ m$^3$ (medium) and 3x$10^6$ m$^3$ (large). These potential avalanche volumes are consistent with the elevations and slopes of the source area. The release area (shown in Figure 3 of the revised paper) was located at an elevation of 5200 m to the north east of the lake following the main axis of the lake."

**Specific Comment 7:**

p. 13: the interface of RAMMS avalanche model, and FLOW3D could be described somewhat more explicitly.

- **Response to Specific Comment 7:**

The RAMMS avalanche model results were not used as direct inputs to the FLOW3D lake model but rather as calibration parameters, the avalanche depth and velocity at the point where the avalanche enters the lake. The FLOW3D lake model was calibrated by adjusting the depth and location of the release area for the fluid representing the avalanche until the depths and velocities of the water entering the lake matched the depths and velocities of the RAMMS avalanche model as it enters the lake.

**Specific Comment 8:**

p. 14, l. 2: I suggest to change conservative risk perspective into worst-case approach

- **Response to Specific Comment 8:**

We use this conservative approach as a synonym of worst-case approach, therefore we accept the suggestion and the change was made accordingly in the document.

**Specific Comment 9:**

p. 16: the interface of FLO-3D and BASEMENT should be described more clearly and in terms of the calibrated parameters. Also, where exactly is BASEMENT started? I think an additional table with the parameters could help.

- **Response to Specific Comment 9:**

[revised manuscript text omitted]

**Specific Comment 11:**

p. 20, l. 7: the area reduction factor could probably also be higher than 20%, considering the building density in Huaraz.

-   **Response to Specific Comment 11:**

We agree with the reviewer, although we have the problem that we don't know which buildings are going to be able to resist the flood so we used this as a best guess. A more detailed study needs to be carried out regarding the effects of inundation of buildings in the city, but this is out of the scope of this work. We think that using *at least* 20% is a valid attempt to represent the obstruction that buildings will impose on the flow which is generally, as far as we know, ignored in most of the models available.

**Specific Comment 12:**

p. 20, l. 23ff: according to Table 2, the intensity matrix for floods and not for debris flows (of the Swiss system) is applied. The model simulates debris flow, so the debris flow intensity levels may be more appropriate.

-   **Response to Specific Comment 12:**

Table 2 in the paper follows the method developed by Garcia et al. (2004: Table 3) which is based on Swiss and Austrian standards (see OFEE et al., 1997; Fiebiger, 1997) modified to fit the results of alluvial fan debris flows in Venezuela. The difficulty arises with the cases h < 0.2 and 0.2 < vh < 1 and h < 0.2 and vh < 0.2, which have shallow water with low velocity. These cases are not covered by the method illustrated in Table 3 of Gacria et al. (2004).

Table 3. Mud and Debris Flow Intensities (from Garcia et al. (2004)).

| Mud or debris-flow intensity | Maximum depth $h$ (m) | | Product of maximum depth $h$ times maximum velocity $v$ ($m^2s^{-1}$) |
|---|---|---|---|
| High | $h > 1.0$ m | OR | $vh > 1.0$ m$^2$s$^{-1}$ |
| Medium | $0.2$ m $< h < 1.0$ m | AND | $0.2 < vh < 1.0$ m$^2$s$^{-1}$ |
| Low | $0.2$ m $< h < 1.0$ m | AND | $vh < 0.2$ m$^2$s$^{-1}$ |

García, R., Rodríguez, J.J., and O'Brien, J.S.: Hazard Zone Delineation for Urbanized Alluvial Fans, Proc. World Water and Environmental Resources Congress, Sehlke, G., Hayes, D.F., and Stevens, D.K. (eds), Salt Lake City, Utah, June 27-July 1, 2004

**Specific Comment 13:**

p. 21, l. 17ff: I see a need to extend how hazard zones were mapped. As mentioned above, a direct conversion of model output to a hazard map is not appropriate (preliminary hazard map may be more appropriate here).

-   **Response to Specific Comment 13:**

We agree with the comment and we have modified the text in the paper accordingly, now from Page 21 line 17 to the dot in line 19 reads as follow:

Old text: "Following Schneider et al. (2014), Raetzo et al. (2002) and Hürlimann et al. (2006) the debris flow intensities have been classified into three classes, and an intensity-likelihood diagram was used to denote three hazard levels (Table 3)."

New text: "Following Schneider et al. (2014), Raetzo et al. (2002) and Hürlimann et al. (2006) the debris flow intensities have been classified into three classes, and an intensity-likelihood diagram was used to denote three preliminary hazard levels (Table 3)."

**Specific Comment 14:**

p.22/23: I suggest to include the results of the comparison with the Heller and Hager model in Table 4. This is of interest.

-   **Response to Specific Comment 14:**

Old Text p.22-23: "As the avalanche impacts the lake, it generates a wave that propagates lengthwise along the lake towards the damming-moraine and attains its maximum height when it reaches the shallow portion at the western end of the lake. Although the wave heights from

FLOW3D are of the same order of magnitude as those calculated from the empirical method (Heller and Hager, 2010), the FLOW3D wave heights are all larger, with the difference in wave heights up to 15% (5.8 m) over the empirically calculated wave height for the large avalanche. Lacking field measurements of lake dynamics or overtopping hydrographs from GLOF events, it is difficult to draw any definitive conclusions about the accuracy of the methods."

New text: "As the avalanche impacts the lake, it generates a wave that propagates lengthwise along the lake towards the damming-moraine and attains its maximum height when it reaches the shallow portion at the western end of the lake. The wave heights are shown in Table 4 for the height of the wave above the moraine crest at the point of overtopping and for the maximum mid-lake wave height. Although the mid-lake wave heights from FLOW3D are of the same order of magnitude as those calculated using the Heller and Hager (2010) method, the FLOW3D wave heights are all larger, with the difference in wave heights up to 13.3% for the large avalanche, and the difference is greater for small and medium avalanches. This may be an indication that the small and medium FLOW3D simulations overestimate the momentum transfer to the lake in the wave-generation process."

Table 4 (in revised paper). Characteristics of Three Avalanche Events of Different Size as Simulated in RAMMS. Overtopping Volume, Flow Rate and Wave Height for Three Avalanche Events as Simulated in FLOW3D for the Current Lake Level and Three Lake Mitigation Scenarios. Comparison of mid-lake wave heights between Heller and Hager (2010) equations and FLOW3D simulations for 0-m lower scenario.

| | Avalanche Event | | |
|---|---|---|---|
| | **Large** | **Medium** | **Small** |
| **Avalanche characteristics in RAMMS** | | | |
| Avalanche size ($10^6$ m$^3$) | 3 | 1 | 0.5 |
| Maximum depth of avalanche material at lake entry (m) | 20 | 15 | 6 |
| Maximum velocity of avalanche material at lake entry (m s$^{-1}$) | 50 | 32 | 20 |
| Time to reach the lake (seconds) | 33 | 36 | 39 |
| % of mass released that reaches the lake in 60 seconds | 84 | 72 | 60 |
| **0 m lower** | | | |
| Overtopping volume ($10^6$ m$^3$) | 1.8 | 0.50 | 0.15 |
| Overtopping peak flow rate (m$^3$s$^{-1}$) | 63,400 | 17,100 | 6,410 |
| Overtopping wave height above artificial dam (m) | 21.7 | 12.0 | 7.1 |
| Maximum mid-lake wave height (m) - Heller and Hager (2010) | 42.2 | 21.1 | 8.8 |
| Maximum mid-lake wave height (m) – FLOW3D | 47.8 | 30.1 | 19.6 |
| **15 m lower** | | | |
| Overtopping volume ($10^6$ m$^3$) | 1.6 | 0.2 | 0.02 |
| Overtopping peak flow rate (m$^3$s$^{-1}$) | 60,200 | 6,370 | 1,080 |
| Overtopping wave height above artificial dam (m) | 38.4 | 27.5 | 25.1 |
| **30 m lower** | | | |
| Overtopping volume (m$^3$) | 1.3 | 0.05 | 0 |
| Overtopping peak flow rate (m$^3$s$^{-1}$) | 48,500 | 1,840 | 0 |
| Overtopping wave height above artificial dam (m) | 60.8 | 42.5 | 0 |

**Specific Comment 15:**

p. 24: I found the evaluation of different lake lowering scenarios particularly useful from an engineering point of view and represents a work that is hardly done.

- **Response to Specific Comment 15:**

The authors appreciate this comment.

**Specific Comment 16:**

p. 25, l. 20ff: I'm not whether failure is the best term here because it may be ambiguous in a case where a breached moraine already exists. I'd rather use full breach development, implying that the lake drains completely. Please clarify this.

- **Response to Specific Comment 16:**

We agree and the text of the paper has been changed:

Old text: "Both the large and medium avalanche events result in a no-failure outcome."

New text: "Both the large and medium avalanche events do result in no breach development."

**Specific Comment 17:**

p. 26, l. 27/28: Almost one hour to cross the urban area seems high to me for a GLOF. Please check whether you may need to adjust the FLO2D model parameters for the urban areas.

- **Response to Specific Comment 17:**

We appreciate the comment and feel that the sensitivity of the travel time to model parameters is important. We have performed a sensitivity analysis on the rheology parameters and the roughness factor in the model and we have added this to the discussion section of the paper after the discussion about the rheology parameter sensitivity (see Response to General Comment 3)

New text: "The model results show that the peak flow takes 55 minutes to cross the city of Huaraz; however, the inundation takes about 45 minutes to cross the city. The inundation spreads through the city diffusing the peak flow and reducing it considerably. Sensitivity analysis showed that increasing the dynamic viscosity, from Glenwood 2 to Glenwood 4, the flow travels faster, arriving at the city 17 minutes earlier, crossing the city in 36 minutes, with the peak flow taking 45 minutes to cross the city. Glenwood 2 and 4 are the lower and higher end, respectively, for the dynamic viscosity parameters used in the sensitivity analysis. When the roughness within the city is reduced to 0.02, the minimum value recommended for asphalt or concrete (0.02-0.05) (FLO2D, 2012) and the 20% area reduction factor is removed, so the flood is limited just by the terrain elevation, the inundation takes 22 minutes to cross the city, 50% of the originally computed time. This value is highly unrealistic since it models the entire land cover of the city as asphalt with no disturbances, buildings, streets, trees, debris, etc.; however, this value can be used as a minimum possible time for the flood to cross the city. If a roughness value of 0.05 is used, then the inundation takes 26 minutes to cross the city. If a value of 0.1 is used, a low but more realistic value, the flood takes 36 minutes to cross the city, and the peak flow takes 43 minutes. Thus, the travel time across the city is more sensitive to changes in roughness values than rheology characteristics. Therefore, we think that 45 minutes for the inundation to cross the city is realistic."

**Specific Comment 18:**

p. 27, l. 1-6: I would be good to also show the arrival times for the small/medium scenarios (cf also Fig. 7).

- **Response to Specific Comment 18:**

We agree with this comment and we include the small and medium scenarios in Table 6. Now the text of Page 26 line 25 after the "." reads:

Old Text: "From the beginning of the avalanche event it takes the flood wave about 1.3 h to reach this location for the large avalanche scenario (Table 6), and the peak flow arrives shortly after. The peak flow takes almost an hour to cross the city. The hydrograph at cross-section 5 shows the discharge in the Rio Santa where the flood exits the city. The peak has attenuated considerably at this point, and arrives after 2.26 h".

New Text: "From the beginning of the large avalanche event it takes the flood wave about 1.3 h to reach cross section 4 (Table 6), and the peak flow arrives shortly after. The peak flow takes about ¾ h to cross the city to cross section 5 and the peak is attenuated by about 50% in the crossing. Values for the medium and small avalanche events are shown in Table 6. They take considerably longer to arrive and cross the city, but their peaks are attenuated about 50% as well.".

Table 6 (in revised paper). FLO2D Simulation Results at Cross-sections Downstream of Lake Palcacocha for the Current Lake Level and a Large Avalanche.

| Cross Section | Avalanche size | Arrival time (hr) | Peak time (hr) | Peak discharge $(m^3s^{-1})$ |
|---|---|---|---|---|
| 1 | Large | 0.05 | 0.05 | 39,349 |
| | Medium | 0.08 | 0.09 | 4,820 |
| | Small | 0.14 | 0.16 | 436 |
| 2 | Large | 0.51 | 0.65 | 3,246 |
| | Medium | 1.07 | 1.14 | 347 |
| | Small | 2.8 | 2.88 | 27 |
| 3 | Large | 0.81 | 0.84 | 2,989 |
| | Medium | 1.67 | 1.71 | 272 |
| | Small | 4.57 | 4.6 | 19 |
| 4 | Large | 1.32 | 1.36 | 1,980 |
| | Medium | 2.9 | 2.97 | 149 |
| | Small | 8.68 | 8.73 | 8 |
| 5 | Large | 2.1 | 2.26 | 920 |
| | Medium | 4.95 | 5.27 | 73 |
| | Small | 15.8 | 16.1 | 4 |

**Specific Comment 19:**

p. 27 (4.6): The decision which scenario to eventually include in a hazard map is also a political and just a scientific question. I would explicitly mention this. To me, the approach taken seems reasonable. We have discussed this issue in a workshop in Huaraz (with participation of Rachel Chisolm, a co-author of this paper). There was not a clear opinion or statement on this. I think assessing the worst-case is something science should do, and its inclusion in terms of a residual hazard zone seems reasonable to me (considering that all hazard zones presented here should be labeled preliminary).

- **Response to Specific Comment 19:**

We agree with this comment and the following text has been added to the paper:

Old text: "The BASEMENT modeling results (see Sect. 4.3.2. hydro-morphodynamic model above) indicate that the overtopping wave generated from the large avalanche event does not cause sufficient erosion to initiate a breach of the moraine and release the lake water, thus rendering a full collapse of the moraine extremely unlikely. The authors consider this scenario nearly impossible given the current understanding of the moraine conditions and the extensive modeling of the moraine using extremely erosive soil characteristics. However, for the sake of providing complete information, the probable maximum flood as a result of a full breach of the damming-moraine at Lake Palcacocha was simulated, assuming this event is the worst possible scenario that could conceivably occur."

New text: "The BASEMENT modeling results (see Sect. 4.3.2. hydro-morphodynamic model above) indicate that the overtopping wave generated from the large avalanche event does not cause sufficient erosion to initiate a breach of the moraine and release the lake water, thus rendering a full collapse of the moraine extremely unlikely. The authors consider this scenario nearly impossible given the current understanding of the moraine conditions and the extensive modeling of the moraine using extremely erosive soil characteristics. The decision which scenario to eventually include in a hazard map is not just a scientific question, but also a political one. The results of the breaching scenario are included since they are needed in order to assess the worst-case scenario, something science and engineering must communicate to the decision makers and stakeholders. However, for the sake of providing complete information, the probable maximum flood as a result of a full breach of the damming-moraine at Lake Palcacocha was simulated, assuming this event is the worst possible scenario that could conceivably occur."

**Specific Comment 20:**

p. 28, l. 21ff: data on past events is available (ie the 1941 GLOF), at least for the downstream mass flow, and this should be discussed, as previously mentioned.

- **Response to Specific Comment 20:**

The authors are not aware of data on the volume of water or debris flow in the 1941 GLOF event.

**Specific Comment 21:**

p. 29, l. 9-11: I agree that the use of a 3D model is adding value to the assessment of lake displacement waves and is likely to capture the complexity better than simpler models. However, I don't quite agree with this statement which seems to me to be overly confident with this model. Overall, there is only limited experience with this kind of model for such environments and there is substantial number of model parameters to be calibrated. I suggest to discuss the uncertainties that are related to this model.

- **Response to Specific Comment 21:**

We agree with the comments on discussion of uncertainty, and we plan on adding a brief discussion of the uncertainties of each process. Sensitivity analysis was included for several of the simulated processes, and although a full presentation of each sensitivity analysis is beyond the scope of this paper, we agree that the paper would be enhanced by a qualitative discussion of the sources of uncertainty and how they influence the robustness of the model.

Old text: "For that reason, it is necessary to represent these processes more fully in simulations and minimize the approximations used in modeling the chain of processes. In this work, this is partially achieved through the use of three-dimensional simulations of lake dynamics and a hydro-morphodynamic model to simulate the damming-moraine erosion process."

New Text: "There is still a considerable amount of uncertainty in the 3D modeling approach for avalanche-generated waves. Nonetheless, even post-event field studies of GLOF waves have difficulty accurately characterizing the wave magnitudes. The 3D modeling approach presented in this paper is intended as an alternative to partially overcome the absence of field data from a GLOF event at the location of the study.

Because field data are not available, we attempted to counteract the inability to calibrate the models by using the best available physical representations in our modeling approach. Because field data were not available at Palcococha, the 3D hydrodynamic model and the hydromorphodynamic model of moraine erosion can give us a better understanding of the likely outcomes of these processes than models that require extensive calibration (e.g., 2D SWE models and breach simulations such as reported in Rivas, et al. (2015)). This is not to say that these models are free from significant uncertainties, but as a model provides better mechanisms to represent the underlying physical phenomena, uncertainties move from the model engine to the physical initial and boundary parameters, reducing the amount of physical or empirical assumptions. Caution is required in any case because lacking a means of calibration/validation, these results represent estimations that might deviate from reality without proper analysis or judgment."

**Specific Comment 22:**

p. 30, l. 6: I suggest to use worst-case instead of conservative approach.

- **Response to Specific Comment 22:**

Response: We agree and we have changed this in the text of the paper.

Old text: "The moraine erosion simulations used an extremely conservative approach, depicting the moraine as a structure with very low erosive resistance."

New text: "The moraine erosion simulations used a worst-case approach, depicting the moraine as a structure with very low erosive resistance."

**Specific Comment 23:**

p. 32, l. 10: I guess you are talking about hazards since the paper does not contain any material on risk.

**Response to Specific Comment 23:**

We agree and the text of the paper has been changed.

Old text: "There is consensus among local authorities, scientists and specialists that Lake Palcacocha represents a GLOF risk with potentially high destructive impact on Huaraz, and this consensus has been validated by the modeling results presented in this paper."

New text: "There is consensus among local authorities, scientists and specialists that Lake Palcacocha represents a GLOF hazard with potentially high destructive impact on Huaraz, and this consensus has been validated by the modeling results presented in this paper."

**Specific Comment 24:**

Figures are of good quality and I particularly like Fig. 7. Table 8 can probably be avoided.

**Response to Specific Comment 24:**

We agree and have deleted Table 8 from the paper.

**HESS-2015-512-Discussions**

Modeling glacial lake outburst flood process chain: the case of Lake Palcacocha and Huaraz, Peru

M. A. Somos-Valenzuela, R. E. Chisolm, D. S. Rivas, C. Portocarrero, and D. C. McKinney

**Response to the Comments of Reviewer 2**

The authors greatly appreciate the insightful and constructive comments of Anonymous Reviewer #2 that helped us to improve the paper.

**General comments**

**General Comment 1:**

Abstract, line 8 and Study area,p.7 line 25. There are different estimations of number of victims during catastrophic GLOF in 1941 exist. For example, Mark Carey (Mark Carey, In the Shadow of Melting Glaciers: Climate Change and Andean Society, 2010, DOI:10.1093/acprof:oso/9780195396065.003.0002) wrote, that "glacial lake outburst flood in 1941 killed 5,000 people and destroyed one-third of the Ancash capital city of Huaraz". So, it is may be better to give several references in one place (for example, in the Study area description).

- **Response to General Comment 1:**

While it is true that Carey (2010) does cite a figure of 5000 deaths as a result of the 1941 Lake Palcacocha GLOF, more recently Wegner (2014) examined the historical records of the Peruvian Red Cross and reports that "According to the final data of the Red Cross itself and the Peruvian government, they calculated, a week after the events, that the dead were around 1800 in addition there were 400 wounded and nearly 1500 families homeless (Peruvian Red Cross, 2004:213)." On page 7, Line 25-27, in fact several references are cited including Carey (2010) and Wegner (2014).

Wegner, S. A.: Lo Que el Agua se Llevó: Consecuencias y Lecciones del Aluvión de Huaraz de 1941, Technical Note 7 of the series "Technical Notes on Climate Change", Ministry of Environment, Lima, Peru, 88 pp., 2014.

Cruz Roja Peruana (2004) Una idea, una acción ; 125 años de la Cruz Roja Peruana. Texto : Carlos Batalla Sotelo. Lima. Tarea Educación Gráfica Educativa. 412 p.

Carey, M.: In the Shadow of Melting Glaciers: Climate Change and Andean Society, Oxford Univ. Press, New York, 2010.

**General Comment 2:**

P.8 Study area. As shown in the fig.1, there are several other river branches with lakes in the area above Huaraz city. Does any possibility of their outburst exist? Or Lake Palcacocha is only one potentially dangerous lake in the basin? It could be interesting to the reader.

- **Response to General Comment 2:**

There are several lakes upstream Huaraz, Lake Palcacocha being the largest, Tullparaju, and Cuchillacocha Lakes being somewhat smaller. We have not done any work on the lakes that the reviewer is referring to except for Palcacocha. Therefore, we cannot report if those lakes are potentially dangerous or not. There is an effort in Huaraz to study those lakes, and three of the coauthors are part of that effort; however, this is out of the scope of this paper and at this point we don't have information to contribute with solid data that support any statement.

**General Comment 3:**

P. 14 Moraine erosion simulation. It is not rare case in the glaciated areas, when moraine dam contains ice or frozen patterns. In such case dam erosion process during outburst flood has other mechanism and erosion can be larger. Whether the damming moraine of Lake Palcacocha may contain ice? This point should be mentioned and discussed.

- **Response to General Comment 3:**

We believe that the Lake Palcacocha terminal moraine does not contain ice, since: (1) there are two structures that reinforce the terminal moraine that are stable and don't show any sign of instability due to ice core melting after several decades of being installed; (2) there is no sign of small collapses on the top of the moraine indicating thermostat activity in the moraine; (3) we have not found ponds formed on the surface of the moraine during our site visits which would indicate melting ice inside; (4) there is no seepage in the surface of the exposed scar of the 1941 GLOF as one walks up from the valley below; and there is no presence of seepage at the toe of the existing moraine. In fact, Vilímek et al. (2005) noted the ponds that have formed in the valley below the toe of the moraine and they report "The location of the ponds along with the observed inflow on their floors favours their seepage origin over pond's formation due to stagnant ice blocks." (Vilímek et al., 2005:111), further "…no evidence proving presence of stagnant ice inside the moraine was found within the Palcacocha Lake area." (Vilímek et al., 2005:112).

Vilímek, V., Zapata, M. L., Klimes, J., Patzelt, Z., and Santillan, N.: Influence of glacial retreat on natural hazards of the Palcacocha Lake area, Peru, Landslides, 2, 107–115, doi:10.1007/s10346- 005-0052-6, 2005.

**General Comment 4:**

P.18. Inundation simulation. FLO-2D model chosen for inundation simulation, doesn't take into account additional erosion and subsequent accumulation of debris during flood wave moving. However, there are several zones of erosion and accumulation of debris from 1941 GLOF along the Paria River, and the same additional erosion and accumulation could be expected for the next GLOF event. Such redeposition is very difficult take into account during modelling, but this model limitation should be mentioned.

- **Response to General Comment 4:**

We agree with the reviewer. There would be considerable erosion and deposition along the river. Flo2D does not estimate erosion when the debris flow module is activated. Additionally, we did not have enough field information to perform such a study. Consequently, we used a prescribed sediment concentration by volume of 50%, which is an upper limit of values recommended in the literature and by the FLO-2D developers (FLO-2D, 2012). We have added discussion of this issue in the revised paper, Page 31 line 12 after the dot:

Old Text: "In this work, a fixed concentration of 50% by volume was used, which is a good upper limit according to the literature, but it may be too high if the solid material available for erosion is not sufficient in the inundation path."

New Text: "A potential GLOF will erode the bank along the river, especially where lateral moraines are present (XS 3), scouring, transporting and depositing soil many times as the flood moves downstream from the lake to the city. Flo2D does not represent this process when using the Mudflow module. Additionally, we did not have field information to perform a study of these effects. Therefore, in this work, a fixed sediment concentration of 50% by volume was used, which is a good upper limit according to the literature and the FLO-2D developers (FLO-2D, 2012), but it may be too high if the material available for erosion is not sufficient in the inundation path."

**General Comment 5:**

P.21 3.6 Hazard identification. To my opinion, it is better to use term "potential hazard" instead "hazard" for described hazard zonation.

- **Response to General Comment 5:**

In the response to Reviewer 1 General Comment 5 we have changed the term "hazard map" to "preliminary hazard map" in recognition that the hazard indicated from modeling results will need to be verified and validated in the field with observations and data. Here, we propose to change the text of the paper to read:

[revised manuscript text omitted]

The relative agreement of the overtopping hydrographs between the BASEMENT and FLOW-3D models shows that it is possible to replicate reasonably well the 3D characteristics of avalanche-generated waves in a 2D SWE model by exaggerating the energy slopes of upstream boundaries (Table 4).

| Page 31: [2] Deleted | Daene McKinney | 4/20/16 5:35:00 PM |
| --- | --- | --- |

Considering the results of the lake lowering mitigation scenarios, the reduction in hazard area in Huaraz is mostly in the high hazard zones (see Table 7). There is a 27% and 45% reduction in the high hazard area (compared to the current lake level) when the lake is lowered 15 or 30 m, respectively.

| Page 50: [3] Deleted | Daene McKinney | 4/15/16 11:15:00 AM |
| --- | --- | --- |

Table 5. Measures of Agreement for Flow Properties at the Overtopping Zone of Lake Palcacocha: Comparing Results from BASEMENT and FLOW-3D Models.

| Flow property | Measure of agreement | Large avalanche |
|---|---|---|
| **Mass** | Peak flow difference (%) | 0.04 |
| | Bias (%) | -17.6 |
| | Normalized Root Mean Squared Error (%) | 24.9 |
| **Momentum** | Peak momentum difference (%) | 7.3 |
| | Bias (%) | -27.3 |
| | Normalized Root Mean Squared Error (%) | 38.8 |
| **Energy** | Grade line slope (fraction per million) | 140 |

Table 6. FLO-2D Simulation Results at Cross-sections Downstream of Lake Palcacocha for the Current Lake Level and a Large Avalanche.

| Cross Section | Arrival Time (hr) | Peak Time (hr) | Peak Discharge ($m^3/s$) |
|---|---|---|---|
| 1 | 0.05 | 0.05 | 39,349 |
| 2 | 0.51 | 0.65 | 3,246 |
| 3 | 0.81 | 0.84 | 2,989 |
| 4 | 1.32 | 1.36 | 1,980 |
| 5 | 2.10 | 2.26 | 920 |

Table 8. Areas of Flood Intensity for Current Lake level, large avalanche and full breaching scenario.

| Low intensity area (km²) | Med. intensity area (km²) | High intensity area (km²) | Total affected area (km²) |
|---|---|---|---|
| 0.31 | 0.40 | 4.22 | 4.93 |

---

## Author Response (AR2)

HESS-2015-512-Discussions

Modeling glacial lake outburst flood process chain: the case of Lake Palcacocha and Huaraz, Peru

M. A. Somos-Valenzuela, R. E. Chisolm, D. S. Rivas, C. Portocarrero, and D. C. McKinney

**RESPONSE TO THE EDITOR DECISION 22-MAY-2016**

**GENERAL COMMENTS**

**General Comment 1:**

**GC-1:** Section 1.2 ("Introduction to glacial lake hazard process chain modeling"), not all mentioned references are explicitly related to the methodology used in the study. It is not clear which achievements of the previous models and methods are utilized, which drawbacks need to be improved in this study and so on.

**Response to GC-1:** The paper has been revised to respond to this comment. In Section 1.2, the unrelated references have been removed and the achievements of previous methods and their drawbacks have been highlighted as well as where and how improvements have been made in this work.

New text in Section 1.2:
"Impulse waves resulting from the impact of an avalanche with the lake were simulated with a three-dimensional hydrodynamic model, FLOW3D (Flow Science, 2012). Much of the work in impulse wave generation, propagation and run-up has been focused on empirical models that replicate wave characteristics based on laboratory observations (Kamphuis and Bowering 1970; Slingerland and Voight, 1979, 1982; Fritz et al., 2004; Heller and Hager, 2010); numerical simulations have been limited to simplified two-dimensional SWE simulations (Rzadkiewicz et al., 1997; Biscarini, 2010; Cremonesi et al., 2011; Ghozlani et al., 2013; Zweifel et al., 2006). However, the 2D SWE representations do a poor job of modeling wave generation and propagation because vertical accelerations are important and cannot be neglected for slide-generated waves (Heinrich, 1992; Zweifel et al., 2006). Analytical calculations of wave run-up and overtopping typically consider simplified lake geometries (e.g., uniform water depth and constant slope of the terminal moraine) that do not necessarily hold true in natural reservoirs (Synolakis, 1987, 1991; Muller, 1995; Liu et al., 2005). The limitations of empirical and 2D SWE models for simulating lake dynamics leave an opening for a more robust approach. Therefore, a fully 3D non-hydrostatic model (FLOW 3D) was used in this paper to simulate the wave generation, propagation and overtopping. "

**GC-2:** Move the following from 3rd (Methodology) Section to Introduction. P.9, line 21 – P.10, line 5 - short concrete review of the lake simulation approaches

**Response to GC-2:** We appreciate the suggestion and we have implemented it by moving the text to the Introduction and merging it with the existing text there.

Revised text in Section 1.2:

"Impulse waves resulting from the impact of an avalanche with the lake were simulated with a three-dimensional hydrodynamic model, FLOW3D (Flow Science, 2012). Much of the work in impulse wave generation, propagation and run-up has been focused on empirical models that replicate wave characteristics based on laboratory observations (Kamphuis and Bowering 1970; Slingerland and Voight, 1979, 1982; Fritz et al., 2004; Heller and Hager, 2010); numerical simulations have been limited to simplified two-dimensional SWE simulations (Rzadkiewicz et al., 1997; Biscarini, 2010; Cremonesi et al., 2011; Ghozlani et al., 2013; Zweifel et al., 2006). However, the 2D SWE representations do a poor job of modeling wave generation and propagation because vertical accelerations are important and cannot be neglected for slide-generated waves (Heinrich, 1992; Zweifel et al., 2006). Analytical calculations of wave run-up and overtopping typically consider simplified lake geometries (e.g., uniform water depth and constant slope of the terminal moraine) that do not necessarily hold true in natural reservoirs (Synolakis, 1987, 1991; Muller, 1995; Liu et al., 2005). The limitations of empirical and 2D SWE models for simulating lake dynamics leave an opening for a more robust approach. Therefore, a fully 3D non-hydrostatic model (FLOW 3D) was used in this paper to simulate the wave generation, propagation and overtopping.

**GC-3:** Move the following from 3rd (Methodology) Section to Introduction. P.11, line 10 – P.13, line 17 - contains concrete review of erosion simulation approaches

**Response to GC-3:** We appreciate the suggestion and we have implemented it by moving the text to the Introduction and merging it with the existing text there.

Revised text in Section 1.2:

[revised manuscript text omitted]

**GC-4:** Move the following from 3rd (Methodology) Section to Introduction. P.15, line 10 – P.15, line 27 - provides review of the inundation simulation methods, etc.

**Response to GC-4:** We appreciate the suggestion and we have implemented it by moving the text to the Introduction and merging it with the existing text there.

Revised text in Section 1.2:

"The resulting lake outburst floods, after breaching or overtopping of the moraine, comprise highly unsteady flows that are characterized by pronounced changes as they propagate downstream (Worni et al., 2012). Calculating downstream inundation caused by a GLOF event requires the simulation of debris flow propagation, since sediment entrainment can cause the volume and peak discharge to increase by as much as three times (Worni et al., 2014; Osti and

Egashira, 2009). One-dimensional models based on the St. Venant equations have been used to model the downstream flood wave propagation of a GLOF, e.g., Klimes et al. (2013) who used HEC-RAS (USACE, 2010) to reproduce the 2010 GLOF from Lake 513 in Peru; Cenderelli and Wohl (2003) who used HEC-RAS to reproduce steady-state aspects of GLOFs in the Khumbu region of Nepal; Byers et al. (2013) who used HEC-RAS to model a potential GLOF from Lake 464 in the Hongu valley of Nepal; Meon and Schwarz, (1993) who used DAMBRK (Fread, 1988) to model a potential GLOF in the Arun valley of Nepal; and Bajracharya et al. (2007) who used FLDWAV (NWS, 1998) to model a potential GLOF from Imja Lake in Nepal. Two-dimensional SWE models are often used to model downstream impacts of GLOFs, e.g., Worni et al. (2012) who used BASEMENT to model flooding from a GLOF at Shako Cho Lake in India; Schneider et al. (2014) who used RAMMS to model debris flow from an overtopping wave at Lake 513 in Peru; Somos-Valenzuela et al. (2015) who used FLO2D to model downstream inundation from a potential GLOF at Imja Lake in Nepal; and Mergili et al. (2011) used RAMMS and FLO2D to simulate flooding from Lake Khavraz in Tajikistan. FLO2D (FLO2D, 2012) is used here to simulate the downstream inundation."

**Specific Comments:**

**SC-1:** In Fig. 11, I have not found areas of different colors, which are related to the aerial images. Please clarify.

**Response to SC-1:** Thank you for pointing this out.  The three RGB channels of the image shown in the legend of the figure have been removed since they are not needed in the figure.

[revised manuscript text omitted]